# The exocyst complex and intracellular vesicles mediate soluble protein trafficking to the primary cilium

S. M. Niedziółka[1,2], S. Datta[1], T. Uśpieński[1,2], B. Baran[1,2], W. Skarżyńska[1], E. W. Humke[3,5], R. Rohatgi [3,4] &
P. Niewiadomski [1✉]

The efficient transport of proteins into the primary cilium is a crucial step for many signaling pathways. Dysfunction of this process can lead to the disruption of signaling cascades or cilium assembly, resulting in developmental disorders and cancer. Previous studies on the protein delivery to the cilium were mostly focused on the membrane-embedded receptors. In contrast, how soluble proteins are delivered into the cilium is poorly understood. In our work, we identify the exocyst complex as a key player in the ciliary trafficking of soluble Gli transcription factors. In line with the known function of the exocyst in intracellular vesicle transport, we demonstrate that soluble proteins, including Gli2/3 and Lkb1, can use the endosome recycling machinery for their delivery to the primary cilium. Finally, we identify GTPases: Rab14, Rab18, Rab23, and Arf4 that are involved in vesicle-mediated Gli protein ciliary trafficking. Our data pave the way for a better understanding of ciliary transport and uncover transport mechanisms inside the cell.

[1] Centre of New Technologies, University of Warsaw, Warsaw, Poland. [2] Faculty of Biology, University of Warsaw, Warsaw, Poland. [3] Department of Medicine, Stanford University School of Medicine, Stanford, CA, USA. [4] Department of Biochemistry, Stanford University School of Medicine, Stanford, CA, USA. [5] Present address: IGM Biosciences, Inc, Mountain View, CA, USA. ✉email: p.niewiadomski@cent.uw.edu.pl

Hedgehog (Hh) signaling is essential for embryonic patterning and organ morphogenesis[1]. Malfunctions of this pathway can lead to developmental disorders and cancer. The expression of Hh target genes is controlled by Gli transcription factors: Gli1 which acts as an activator, and Gli2/Gli3, which displays both activator and repressor functions[2].

Processing of Gli transcription factors to activator and repressor forms requires their efficient transport to the primary cilium, which integrates proteins necessary to a variety of Gli modifications[3–8]. Cilia are indispensable for the transduction of the Hh signal and the translocation of Gli activators into the nucleus[9]. In humans, defects in the ciliary function and the trafficking of ciliary proteins often result in developmental defects associated with the dysfunction of the Hh/Gli cascade.

Gli proteins are large and slowly diffusing proteins, so it is puzzling how they accumulate at the cilium within a mere few minutes upon signal reception[10]. This accumulation is a result of a three-step process: (1) targeted transport to the cilium base, (2) gated entry through a diffusion barrier, and (3) active trafficking along the cilium. The mechanisms of Gli transition across the diffusion barrier and the model of transport from the base to the tip are relatively well-described[11–13]. Specifically, inside the cilium the Gli proteins appear to utilize the conserved intraflagellar transport (IFT) system for the delivery from the base of the cilium to the tip and back to the base[10,14–18]. At the base of the cilium, Gli proteins must pass the diffusion barrier reminiscent of the nuclear pore with the help of some of the same molecular players as those involved in nucleocytoplasmic shuttling: Ran, importins, and exportins[12,19,20]. However, it is still unclear how Gli proteins are delivered from the cytoplasm to the cilium base in the first place.

Most previous studies on protein delivery to the ciliary base were focused on membrane proteins. Three different transport routes have been described for their delivery from the Golgi complex to the primary cilium[21]. Some ciliary proteins first reach the plasma membrane and then move to the ciliary membrane by lateral transport[22]. Others reach the base of the cilium using regulated vesicular transport, either directly or through the recycling trafficking pathway[23].

The process of protein trafficking to the primary cilium is supported by many players involved in endocytosis and the vesicle transport machinery[24–26]. Prominent among them are small GTPases, which act as molecular switches that allow for the guidance of their associated vesicles[27–29]. In addition to GTPases, the protein ciliary trafficking depends on several multiprotein complexes, such as the BBsome[30,31] and the exocyst[32,33]. The exocyst, in particular, is a conserved protein complex that mediates the tethering of secretory vesicles to the plasma membrane[34]. Many protein trafficking routes depend on the exocyst, including those for protein delivery to the cell junctions and protein endocytosis and recycling[35,36]. The transport function of the exocyst relies on its interaction with membrane phospholipids[37,38], polarity proteins[39,40] and small GTPases[40–43]. Importantly, the exocyst interacts with the ciliary transport machinery, and has been implicated in the transport of transmembrane proteins necessary for ciliogenesis and signaling[23,42,44–47], but its role in the trafficking of soluble ciliary proteins has not been demonstrated.

In our quest to identify the molecular machinery that delivers Gli proteins to the cilium base, we performed a proteomic analysis of Gli3 interactors. Interestingly, among Gli3-binding proteins, we detected several exocyst subunits[34]. Loss-of-function assays show the dependence of Gli2 and Gli3 ciliary localization on the exocyst. Consistent with the role of this complex in vesicle trafficking, we show that Gli2 uses intracellular vesicles as trafficking vehicles. In addition, several small GTPases, including Rab14, Rab18, Rab23, and Arf4, regulate the ciliary transport of Gli2. Finally, we show that this vesicle-based transport machinery is used for the ciliary delivery of Lkb1, another soluble protein that concentrates at cilia. Our study uncovers a vesicle-dependent transport pathway for soluble ciliary proteins and sheds new light on the mechanisms of protein delivery to the cilium base.

## Results

**The exocyst complex interacts with Gli3.** To identify proteins that help guide Gli proteins to the primary cilium, we immunoprecipitated proteins that interact with Gli3 in cells treated with the Smoothened (Smo) agonist SAG[48]. Cells were separated into "nuclear" and "cytoplasmic" fractions using hypotonic lysis and low-speed centrifugation, and each fraction was immunoprecipitated with anti-Gli3 antibodies. The eluates were separated using SDS-PAGE, and prominent bands were submitted for MS-based protein identification (Fig. 1a).

Mascot's search of the MS spectra yielded a total of 898 unique proteins. From this dataset, we selected 473 high confidence Gli3 interactors by rejecting proteins that are frequently found as IP/MS contaminants based on the CRAPome database[49]. In this dataset, we found well-known Gli interaction partners, such as SuFu, Kif7, and Xpo7[50–53]. The dataset was enriched for proteins involved in intraciliary and vesicle transport, chromatin remodeling, and DNA repair (Fig. 1b) and contained components of multi-subunit ciliary transport complexes, including the BBSome and the exocyst (Fig. 1c, Supplementary Data 1).

Because exocyst, a multi-subunit protein complex involved in vesicle transport and docking[54], had previously been implicated in the trafficking of proteins to primary cilia, we decided to focus on its components as potential mediators of the Gli proteins delivery to the cilium base. The exocyst has mostly been studied in the context of its binding to intracellular vesicles and the plasma membrane, but the subunits that were specifically enriched in the Gli3 interactome are positioned away from the putative lipid-binding surface of the complex, consistent with Gli3 being a soluble, rather than a lipid-embedded protein (Fig. 1d).

In agreement with the proteomic data, Gli3, as well as Gli2, co-immunoprecipitate with Sec5 and Sec3 (Fig. 2a, S1a). Moreover, Sec5 and Gli2 tightly colocalize in cells, as shown using the proximity ligation assay (Fig. 2b, Supplementary Fig. S1b). Similarly, overexpressed Sec3, Sec5, and Sec8 interact with the constitutively active Gli2 mutants Gli2(P1-6A) (Fig. 2c, d)[55]. We decided to use Gli2(P1-6A) in most experiments because it localizes to cilia in the absence of upstream activation, allowing us to study its trafficking independently of the transport of membrane proteins regulating endogenous Gli proteins, such as Smo and Ptch[56,57].

To identify the Gli2 domain responsible for interaction with the exocyst, we performed co-immunoprecipitation of Sec3/5/8 with the N-terminal domain of Gli2 and a construct lacking the N-terminus. The exocyst subunits interact with the N-terminus of Gli2 (HA-Gli2-N) but interact only weakly with Gli2(P1-6A)-ΔN (Fig. 2e, f).

**Trafficking of Gli2 to cilia depends on the exocyst.** Because the exocyst is required for the trafficking of some ciliary proteins, we hypothesized that the loss-of-function of the exocyst could impair Gli ciliary localization at the tip of primary cilia. To test this assertion, we knocked down individual exocyst subunits in cells expressing Gli2(P1-6A), in which the HA-tagged Gli2 mutant accumulates at cilia tips (Supplementary Fig. S2a). Both shRNA- (Fig. 3a) and siRNA-mediated knockdown (Fig. 3b) of exocyst subunits resulted in a significant reduction of Gli2(P1-6A) ciliary

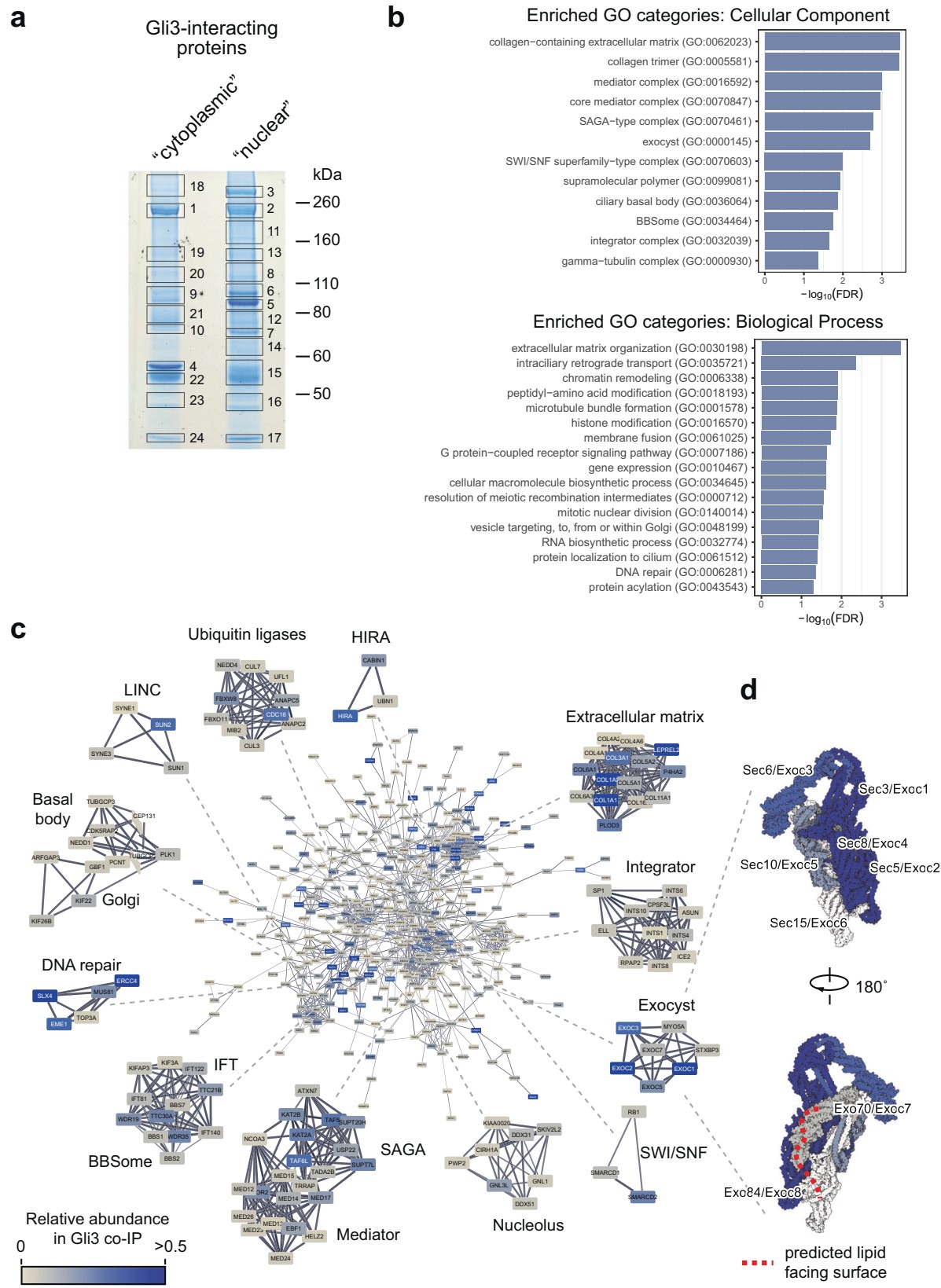

localization in NIH/3T3 (Fig. 3c, d) and mIMCD3 cells (Supplementary Fig. S2b). Similarly, the SAG-induced ciliary accumulation of endogenous Gli2 is reduced in NIH/3T3 cells with Sec3/5/8 knockdown (Supplementary Fig. S2c). Importantly, exocyst loss of function does not reduce the total number of ciliated cells, suggesting that it does not have a broad effect on ciliogenesis under experimental conditions that we used (Supplementary Fig. S2d).

Similarly, mislocalization of Sec5 using the mitochondrial trap[58] impairs the ciliary trafficking of Gli2(P1-6A). We fused

**Fig. 1 Gli3 interactome is enriched for proteins involved in ciliary transport of vesicles. a** NIH/3T3 Flp-In cells were treated with 100nM SAG for 4h, roughly fractionated into a "cytoplasmic" and "nuclear" fraction, and each fraction was pulled down using magnetic beads coated with the anti-Gli3 antibody. Proteins were resolved on SDS-PAGE, the gel was stained with coomassie brilliant blue and prominent bands were excised for mass spectrometry-based protein ID. Shown is the image of the coomassie-stained gel with each of the excised bands indicated and numbered. Gli3 is enriched in bands 1 ("cytoplasmic") and 2 ("nuclear"). **b** MS-identified proteins from all bands were pooled and common MS-AP contaminants (>10% FDR from the CRAPome database[49]) were removed. PANTHER[114] was used to find overrepresented Gene Ontology (GO) terms in the "PANTHER GO—Slim Biological Process" and "PANTHER GO—Slim Cellular Component" categories. Top-level enriched GO terms are shown with their corresponding $-\log10$(FDR) values. **c** High confidence Gli3 interactors identified by MS were connected into a network using the STRING[118] plugin in Cytoscape. Shown is the main protein network with the node color representing the approximate relative abundance of the protein in the Gli3 interactome and the edge thickness corresponding to the confidence of connection between proteins in the STRING database. Also shown are highly interconnected sub-networks identified using MCODE clustering, which typically corresponds to protein complexes or multiprotein functional units. **d** The exocyst complex structure (PDB ID: 5yfp[34]) was rendered using Illustrate[119] with each subunit colored according to its abundance in Gli3 IP/MS as in c Subunits not identified in our experiment are rendered in white. The red dashed line corresponds to the predicted surface of the exocyst complex that comes into contact with the plasma membrane lipids[34]. Each subunit is labeled with its alternative gene names.

---

Sec5 with the mitochondrial protein Tom20 and mScarlet and co-expressed the resulting Tom20-mScarlet-Sec5 construct with Gli2(P1-6A) (Fig. 4a). We observed a reduced Gli2 ciliary level in cells overexpressing the Tom20-mScarlet-Sec5 mitochondrial trap, compared to those overexpressing two negative control constructs—Tom20-mScarlet and mScarlet-Sec5 (Fig. 4b, S2e).

Finally, the exocyst inhibitor endosidin2[59] reduces Gli2(P1-6A) ciliary localization in the stable cell line after just two hours of treatment (Fig. 4c).

Because the exocyst binds to Gli2 mostly via its N-terminal domain (Fig. 2e, f), we suspected that removing the N-terminus would impair Gli2 ciliary accumulation. Accordingly, we observed a strong reduction of the Gli2(P1-6A)-ΔN mutant localization in the primary cilium compared to the full-length protein (Fig. 4d).

Having demonstrated that the exocyst is required for the trafficking of Gli to cilia, we wondered if the localization of the exocyst is affected by Hh pathway activation. Indeed, the treatment with SAG increases the amount of Sec3 and Sec5 at the ciliary base (Fig. 4e).

**Gli2 associates with intracellular vesicles.** While the best-known role of the exocyst complex is the transport of vesicle-embedded membrane proteins, our results suggest that soluble cytoplasmic Gli proteins may also use the exocyst as a vehicle for intracellular trafficking. We, therefore, wondered if Gli proteins, like membrane receptors, use vesicles for their transport into the cilium. To verify this hypothesis, we used super-resolution AiryScan microscopy to image cells co-expressing HA-Gli2(P1-6A) and EGFP-Sec5, and surprisingly, we observed Gli2 around Sec5-positive vesicle-like structures. It suggests that Gli2 could accumulate on the surface of vesicles, where it could interact directly with the exocyst (Fig. 5a). Similarly, Gli2 associates with distinct Sec3-positive structures throughout the cell (Fig. 5a).

We also looked at Gli2 localization by immunogold electron microscopy. In HEK293T cells overexpressing EGFP-Gli2(P1-6A), we observed EGFP-positive clusters adjacent to membrane vesicle-like structures (Fig. 5b).

To check if Gli-positive structures represent intracellular vesicles, we isolated vesicles using cell fractionation. HA-Gli2(P1-6A), endogenous Gli3, and Sec5 co-fractionated with the endosome marker EEA1 in the endosomal fraction. ERK was used as the cytoplasmic control marker. The total abundance of proteins in fractions we showed by silver staining (Fig. 5c, S3a).

The most likely explanation for our results is that Gli proteins are transported on the surface of vesicles towards the ciliary base. We reasoned that the vesicles likely fuse at the base of the primary cilium[60] releasing Gli proteins. Therefore, for continuous trafficking, new vesicles that would carry Gli proteins would

need to form. The two potential sources of these vesicles are the Golgi apparatus via the exocytic pathway[61,62] and the plasma membrane by endocytosis[63–65]. We inhibited endocytosis using two inhibitors: dynasore[66] and pitstop2[67] in cells expressing constitutively active Gli2. Surprisingly, after 2h of dynasore treatment, we observed an almost complete inhibition of Gli2 ciliary accumulation (Fig. 5d). This effect was independent of Smo because treatment with two Smo inhibitors cyclopamine and vismodegib did not affect the Gli2(P1-6A) ciliary level (Fig. 5e and Supplementary Fig. S3b–g).

If the dynasore effects are a consequence of the reduced rate of new vesicle formation, we would expect these effects to be fully reversible once the proper formation of vesicles is restored. We used a pulse-chase assay with 2h vismodegib + dynasore treatment, and then we washed out dynasore from the media and collected cells at several time points. We observed a clear recovery of Gli2 ciliary transport within 1h from the dynasore washout (Fig. 5d). This effect was present in both NIH/3T3 cells (Fig. 5d) and in IMCD3 cells (Supplementary Fig. S3f), suggesting that cells utilizing both the intracellular and extracellular ciliogenesis pathways (NIH/3T3 and IMCD3, respectively[68]) use endocytic vesicles to deliver Gli proteins to cilia.

As an alternative to dynasore, we also used another commercial clathrin-dependent endocytosis inhibitor pitstop2. Because of its lethal effect on NIH/3T3 fibroblasts in less than 30min, we treated cells with pitstop2 for 15 min, followed by a 30 min incubation without the drug. Similar to the dynasore effects, we observed a decrease of Gli2 ciliary level in pitstop2-treated cells (Fig. 5f). Importantly, both dynasore and pitstop2 were effective at reducing endocytosis at the doses and times used (Supplementary Fig. S3c) and neither inhibitor induced gross morphological changes indicative of toxicity under these conditions (Supplementary Fig. S3d, g).

To determine if the vesicle transport from the cis-Golgi was also important for Gli2 ciliary trafficking, we treated stable HA-Gli2(P1-6A) cells with brefeldin A, a Golgi-disrupting drug[69]. We did not observe changes in Gli2 ciliary localization after 2h treatment (Fig. 5g).

The stimulation of target gene transcription by Gli2 is enhanced by its localization at the cilium[9,70]. We expected that dynasore would inhibit Hh target gene transcription in cells stably expressing the Gli2(P1-6A). Indeed, the expression of the Hh target gene Gli1 was decreased after dynasore treatment, although the expression of HA-Gli2(P1-6A) was unchanged (Fig. 5h).

**Rab and Arf proteins mediate Gli2 transport.** The trafficking of vesicles in cells is guided by the reversible association of small GTPases, especially from the Rab and Arf families[25,71]. Because

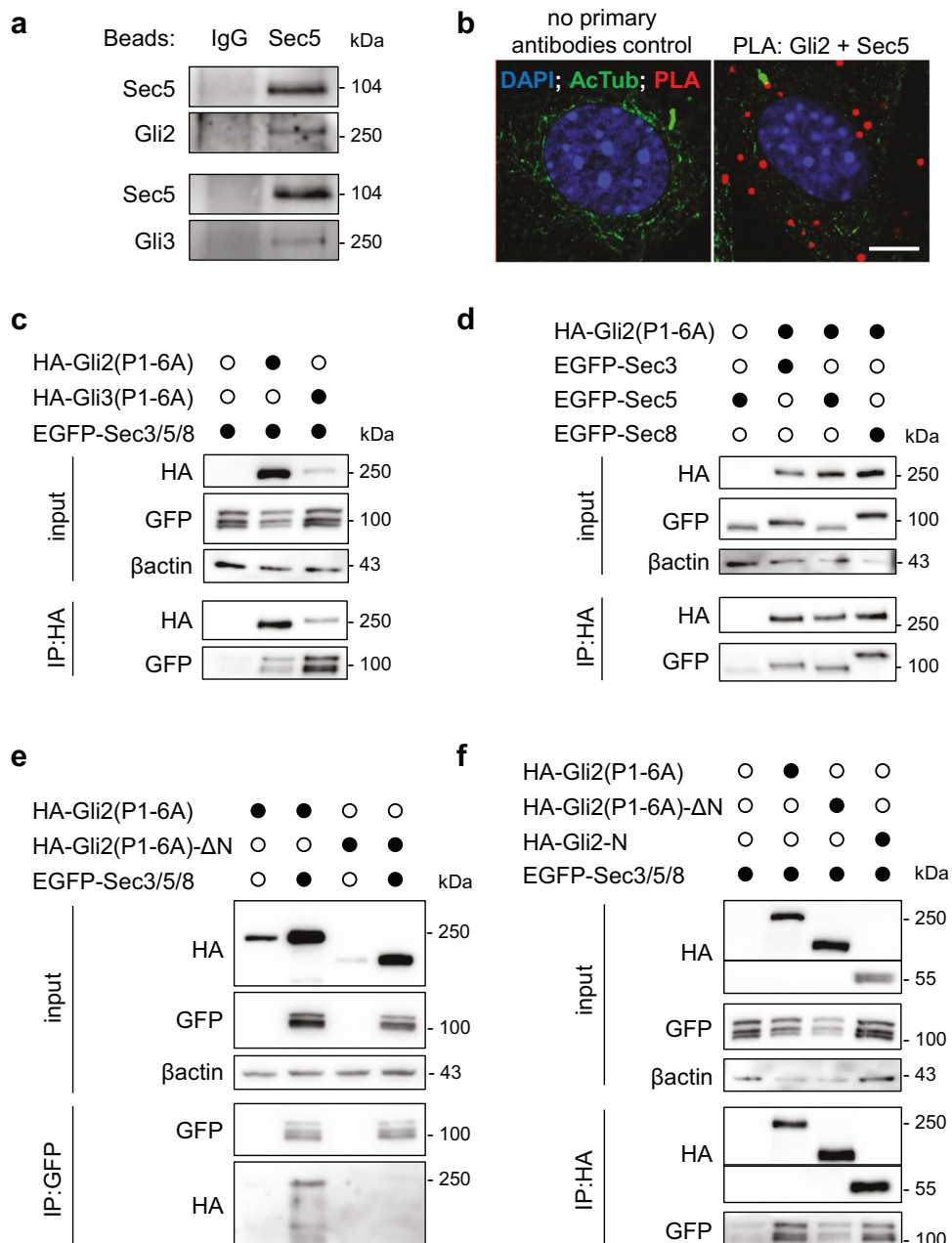

**Fig. 2 Exocyst subunits interact with Gli2 and Gli3. a** Co-immunoprecipitation of endogenous Sec5 with Gli2 and Gli3. Beads were coated with anti-Sec5 antibodies. Rabbit IgG was used as a control. **b** Proximity Ligation Assay with anti-Gli2 and anti-Sec5 antibodies in NIH/3T3 mouse fibroblasts. Sites of interaction are marked in red. Cilia were stained with anti-acetylated tubulin (green), and nuclei with DAPI (blue). Scale bar 5μm. **c** Co-immunoprecipitation of overexpressed HA-Gli2(P1-6A) and HA-Gli3(P1-6A) with the exocyst subunits Sec3, Sec5, and Sec8 tagged with EGFP in HEK293T cells using anti-HA beads. **d** Co-immunoprecipitation of overexpressed HA-Gli2(P1-6A) with single exocyst subunits Sec3, Sec5 and Sec8 tagged with EGFP in HEK293T cells using anti-HA beads **e** Reciprocal co-immunoprecipitation of overexpressed EGFP-tagged Sec3, Sec5 and Sec8 with HA-Gli2(P1-6A) constructs using anti-GFP beads. **f** Co-immunoprecipitation of overexpressed HA-Gli2(P1-6A) truncation constructs with the exocyst subunits Sec3, Sec5, and Sec8 tagged with EGFP in HEK293T cells using anti-HA beads.

their association with vesicles and associated proteins is transient, we hypothesized that under the stringent conditions of our initial co-IP/MS, the Gli-associating GTPases may have been washed away from the bait protein. Thus, we performed another co-IP/MS, with less stringent detergents, using HA-Gli2(P1-6A) as bait in cells that either had normal cilia or were devoid of cilia by means of overexpression of a dominant-negative mutant Kif3a motor[72]. We expected the GTPases promoting Gli ciliary trafficking to be associated with Gli2 in ciliated, but not in unciliated cells (Fig. 6a).

We identified 200 high-confidence interactors (<10% FDR in the CRAPome database) including the same well-known regulators of Gli, such as SuFu, Kif7, Xpo7, and Spop[51,53,73,74], as well as component proteins of the cilium and basal body (Fig. 6b, Supplementary Data 2). Among proteins associated with Gli2(P1-6A) in ciliated cells were Rab14, Rab5c, Rab11b, Rab18, and Arf4 (Fig. 6c). In addition, we tested two other Rab-family GTPases: the well-known Hh regulator Rab23[75–77] and Rab8, which cooperates with the exocyst in the trafficking of membrane receptors to primary cilium[43,78].

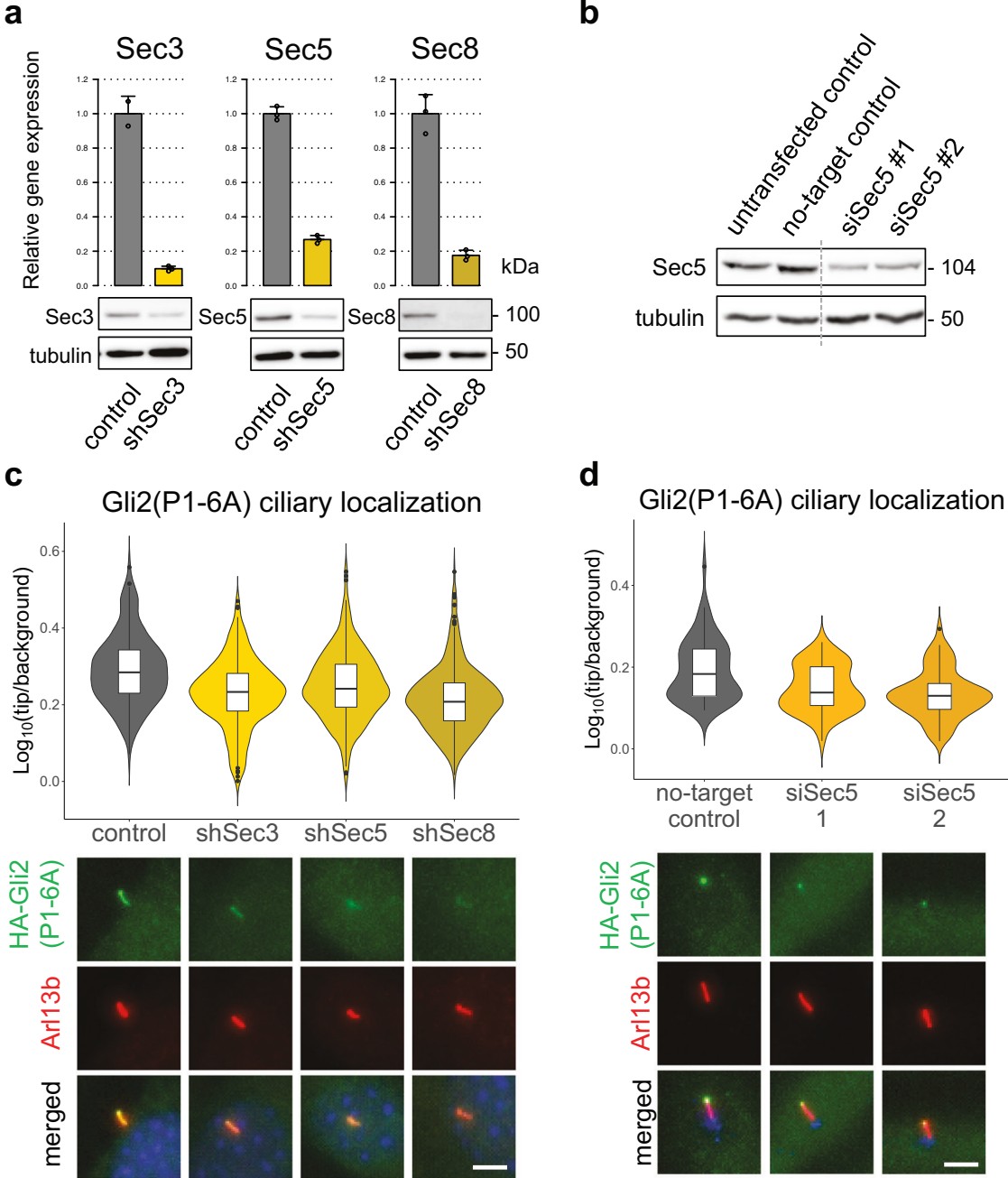

**Fig. 3 Knockdown of exocyst subunits decreases Gli2 ciliary localization. a** mRNA expression levels of the indicated genes in cells stable expressing Gli2(P1-6A) and transduced with shRNA against each of the genes were measured using qRT-PCR. Control cells were transduced with shRNA against luciferase. The protein level of the indicated proteins was detected by western blot. **b** The protein level of Sec5 in cells transfected with siRNA against Sec5 or non-targeting control siRNA. **c** Relative localization at the cilium tip of stably expressed Gli2(P1-6A) in cells with shRNA knockdown of Sec3, Sec5, and Sec8. Results are presented as violin plots of log-transformed ratios of fluorescence intensity of anti-HA staining at cilia tips to the intensity in the surrounding background. Cilia per variant $n > 170$. Student's $t$ test analysis control-shSec3 $p$-value = 5.399e-12; control-shSec5 $p$-value = 2.206e-06; control-shSec8 $p$-value < 2.2e-16. Representative images of Gli2(P1-6A) ciliary localization for each condition are presented below. Arl13b was used as a ciliary marker. **d** Relative localization at the cilium tip of Gli2(P1-6A) in cells transfected with indicated siRNAs. Fluorescence intensities were quantified as in Fig. 3c from $n > 60$ cilia per group. Student's $t$ test for no-target control-siRNA2 $p$-value = 0.0001015; for no-target control-siRNA3 $p$-value = 1.581e −06. Representative images of Gli2(P1-6A) ciliary localization for each condition are presented below. Arl13b was used as a ciliary marker and pericentrin (blue) as a basal body marker. Scale bars 5 μm.

Initially, we established by co-IP that Rab14, Rab18, Rab23, and Arf4 proteins interact with Gli2(P1-6A) (Fig. 7a). In contrast, two Rab GTPases that had been implicated in ciliary trafficking of membrane proteins: Rab8 and Rab11a, do not strongly bind to Gli2(P1-6A) (Supplementary Fig. S4a).

Subsequently, we performed loss-of-function experiments using shRNA and CRISPR/Cas9 mutagenesis. The knockdown of Rab14, Rab18, and Arf4 caused the reduction of the Gli2(P1-6A) ciliary level (Fig. 7b–d). Likewise, the CRISPR/Cas9-mediated Rab14, Rab18, Rab23, and Arf4 knockout, but not that of Rab8 or

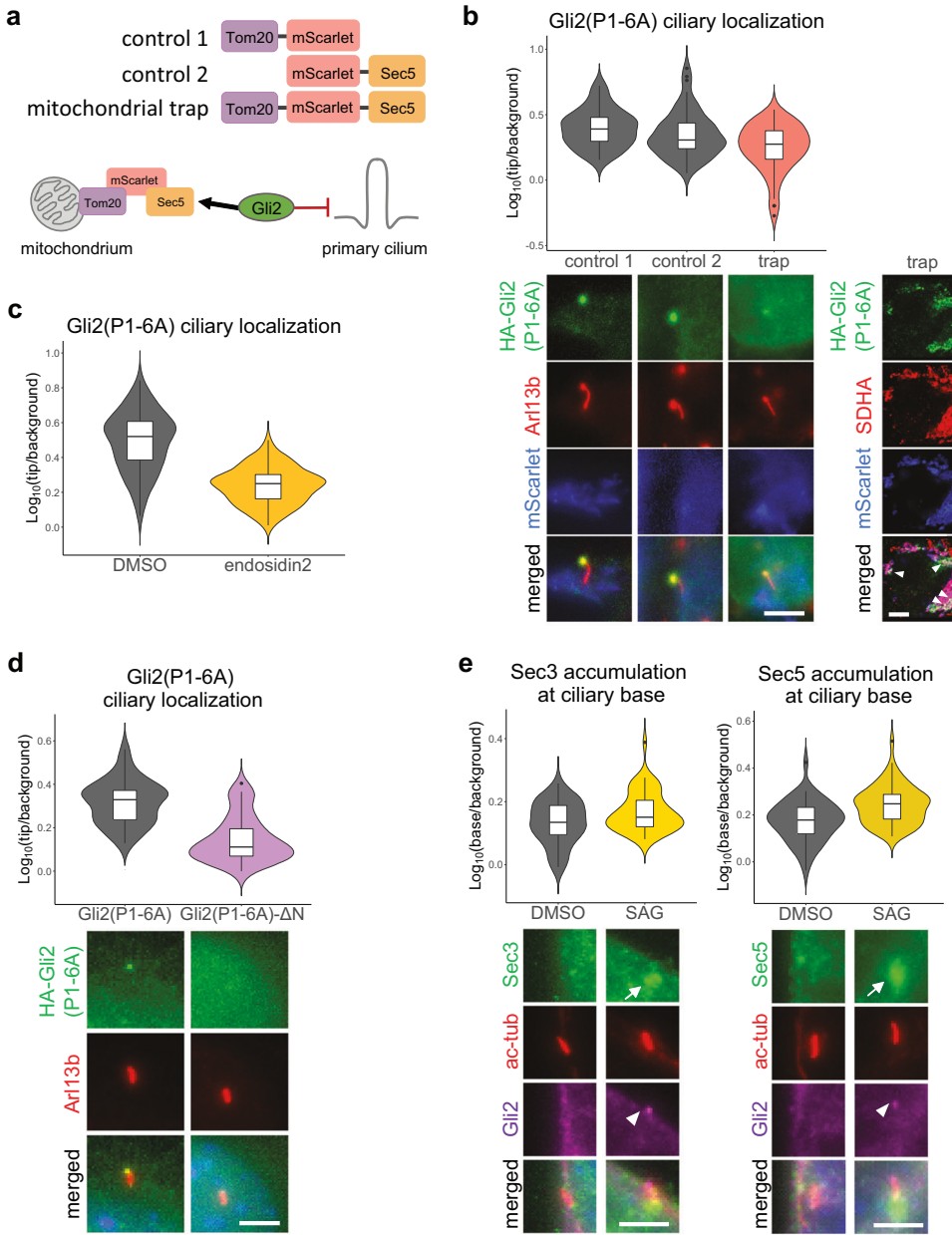

**Fig. 4 Impairment of exocyst function reduces Gli2 ciliary localization. a** Schematic representation of the exocyst mitochondrial trap constructs (top) and assay (bottom). **b** Relative localization at the cilium tip of Gli2(P1-6A) in HEK293T cells co-transfected with the HA-Gli2(P1-6A) and the indicated constructs in $n > 40$ cilia. Fluorescence intensities were quantified as in Fig. 3c. Student's test for control 1 vs trap $p$-value = 4.28e−06; control 2 vs. trap $p$-value = 0.002. Representative images of Gli2(P1-6A) ciliary localization are presented below. Lower magnification images showing overlap of Gli2 with mScarlet and a mitochondrial marker SDHA are shown on the right (arrowheads show colocalization). **c** The exocyst inhibitor endosidin2 blocks the ciliary accumulation of Gli2(P1-6A). Relative localization at the cilium tip of Gli2(P1-6A) in NIH/3T3 cells expressing HA-Gli2(P1-6A) treated for 2h with DMSO or 200μM endosidin2 was measured in $n > 100$ cilia per group. Fluorescence intensities were quantified as in Fig. 3c. Student's $t$ test $p$-value < 2.2e−16. **d** Gli2(P1-6A)-ΔN is largely excluded from the tip of cilia. Relative localization at the cilium tip of Gli2 constructs stably expressed in NIH/3T3 cells. Fluorescence intensities were quantified as in Fig. 3c in $n > 50$ cilia per group. Student's $t$ test $p$-value = 2.2e−14. Representative images are presented below. Arl13b was used as a ciliary marker. **e** Effect of Smoothened agonist (SAG) treatment (24h; 200nM) on the accumulation of Sec3 and Sec5 at the ciliary base in NIH/3T3 cells. Cells were stained with anti-Sec3 or anti-Sec5 and the ciliary marker acetylated α-tubulin (ac-tub). Relative localization at the cilium base was measured in $n > 40$ cells per group as in Fig. 3c. Student's $t$ test Sec3: control vs SAG $p$-value = 0.005; Sec5: control vs SAG $p$-value = 4.6e −05. Representative images for each condition are presented below. White arrows show Sec3/5 accumulation and white arrowheads show Gli2 ciliary accumulation. Scale bars 5μm.

Rab11a, also significantly decreased the Gli2(P1-6A) ciliary accumulation (Fig. 7e, S4b). Moreover, we engineered cell lines expressing dominant-negative Rab23$^{S51N}$ and Arf4$^{T31N}$ mutants from doxycycline-inducible promoters. Consistent with shRNA-and CRISPR/Cas9-based experiments, we observed a significant decrease of Gli2(P1-6A) ciliary accumulation in cells expressing Arf4 and Rab23 mutants, but not those of Rab8 or Rab11a (Fig.7f, S4c).

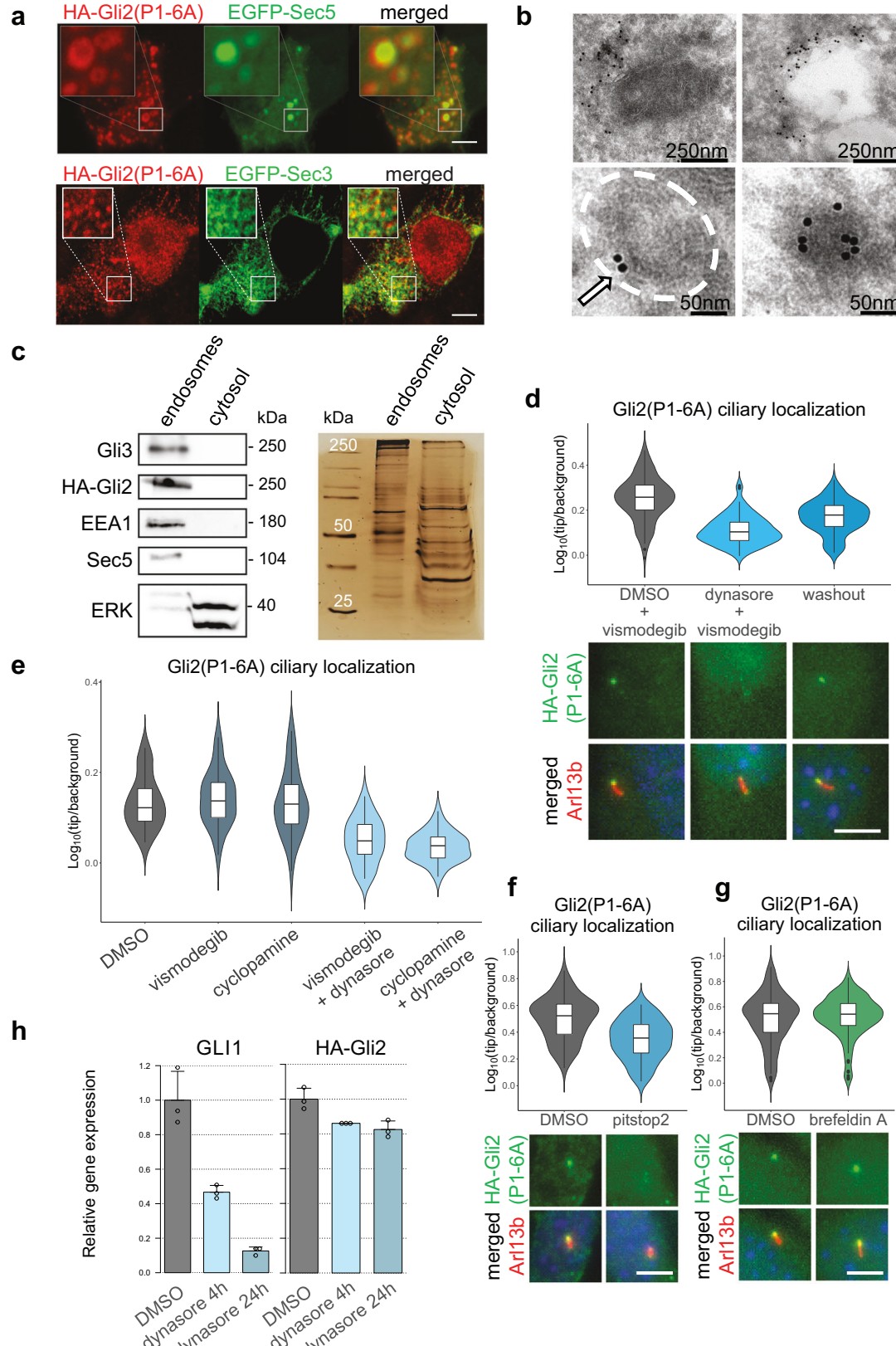

**The trafficking of Lkb1, but not Ubxn10, depends on endocytosis and the exocyst**. We wanted to know if the mechanism of transport to cilia with the use of endocytic vesicles was unique to Gli proteins or more common among other soluble ciliary proteins. For this purpose, we imaged several HA or GFP tagged soluble ciliary candidate proteins: HA-Dvl2[79], Kap3a-EGFP[80],

HA-Lkb1[81], HA-Mek1[82], HA-Nbr1[83], HA-Raptor[84], Tbx3-GFP[85], and Ubxn10-GFP[86]. Only two proteins clearly localized at cilia in NIH/3T3: Ubxn10-GFP, and HA-Lkb1 (Fig. 8a and Supplementary Fig. S5).

To examine if the ciliary serine-threonine kinase Lkb1 uses an analogous transport mechanism, we treated stable expressing

**Fig. 5 Gli2 associates with intracellular vesicles. a** Airyscan fluorescence imaging of HEK293T cells co-transfected with HA-Gli2(P1-6A) and EGFP-Sec5 or EGFP-Sec3 stained with anti-HA. Insets show high magnification of the Sec5/3- and HA-Gli2(P1-6A)-positive structures. **b** Electron microscopy images of HEK293T cells transfected with EGFP-Gli2(P1-6A) and labeled with immunogold-conjugated anti-GFP. EGFP-positive signal accumulates around vesicle-like structures. **c** Cells stably expressing HA-Gli2(P1-6A) were fractionated using the endosome isolation kit and the fractions were resolved using SDS-PAGE. Immunoblot shows HA-Gli2(P1-6A), Gli3, and Sec5 in the endosomal fraction. EEA1 was used as a marker of the endosomes, and ERK was used as a cytosolic fraction marker. The same protein samples were resolved by SDS-PAGE and the gel was silver-stained, showing similar total protein abundance in both fractions. **d** Dynasore impairs Gli2(P1-6A) ciliary localization. Cells were treated with vismodegib in the presence or absence of dynasore for 2h hours and then the drugs were washed out for 1h. Relative localization of Gli2(P1-6A) at the cilium tip was measured as in Fig. 3c for $n > 80$ cilia per group. Student's $t$ test DMSO+vismodegib vs dynasore+vismodegib $p$-value $< 2.2e{-}16$; dynasore+vismodegib vs washout 1h $p$-value $= 4.25e{-}05$. Representative images of Gli2(P1-6A) ciliary localization for each condition are presented below. Arl13b was used as a ciliary marker. **e** Effect of dynasore treatment on Gli2(P1-6A) ciliary accumulation. NIH/3T3 cells with stable expression of HA-Gli2(P1-6A) were treated with dynasore (4h; 40μM) in the presence of Smo inhibitors vismodegib (4h; 3μM) and cyclopamine (4h; 10μM). The Smo inhibitors were used to ensure that the effect of dynasore was not due to its influence on Smo or Ptch trafficking. The Smo inhibitors did not influence Gli2(P1-6A) ciliary accumulation, as expected, and did not prevent dynasore from inhibiting Gli2(P1-6A) localization at the cilium tip. Relative localization of Gli2(P1-6A) at the cilium tip was measured as in Fig. 3c for $n > 30$ cilia per group. Student's $t$ test DMSO vs vismodegib $p$-value $= 0.5533$; DMSO vs vismodegib+dynasore $p$-value $= 9.047e{-}08$; DMSO vs cyclopamine $p$-value $= 0.8634$; DMSO vs cyclopamine+dynasore $p$-value $= 1.708e{-}10$. **f** Effect of pitstop2 treatment on Gli2(P1-6A) ciliary accumulation. Pitstop2 (30μM) was used for 10 min and then washed out to avoid its toxicity. Effect of treatment was observed 30 min after washout. Relative localization of Gli2(P1-6A) at the cilium tip was measured as in Fig. 3c for $n > 80$ cilia per group. Student's $t$ test DMSO vs pitstop2 30 min washout $p$-value $= 2.486e{-}10$. Representative images of Gli2(P1-6A) ciliary localization for each condition are presented below. Arl13b was used as a ciliary marker. **g** Effect of brefeldin A treatment on Gli2(P1-6A) ciliary accumulation. Cells were treated with DMSO or brefeldin A (5 μg/ml) for 2h. Relative localization of Gli2(P1-6A) at the cilium tip was measured as in Fig. 3c for $n > 140$ cilia per group. Student's $t$ test DMSO vs brefeldin A $p$-value $= 0.4565$. Representative images of Gli2(P1-6A) ciliary localization for each condition are presented below. Arl13b was used as a ciliary marker. **h** The relative mRNA expression level of Gli1 (Hh pathway activity marker), and HA-Gli2(P1-6A) after 4h and 24h of dynasore treatment. Scale bars for immunofluorescence images 5μm.

HA-Lkb1 cells with dynasore and observed decreased Lkb1 ciliary level (Fig. 8b). Similar to Gli2, ciliary accumulation of HA-Lkb1 also dropped after the shRNA knockdown of Sec3/5/8 (Fig. 8c). Accordingly, we detected HA-Lkb1 in the endosomal fraction (Fig. 8d). Finally, we observed using co-IP that Lkb1 binds to the exocyst subunits (Fig. 8e).

Another soluble ciliary protein that we studied was Ubxn10. Dynasore treatment did not negatively affect the ciliary trafficking of Ubxn10-GFP (Fig. 8f). Unlike for Gli2(P1-6A), we observed no effect of Sec5 knockdown on Ubxn10 ciliary localization (Fig. 8g). Consistent with these results, we detected Ubxn10 predominantly in the cytosolic cell fraction (Fig. 8h).

## Discussion

The cilium is an essential organelle that relays environmental signals to the nucleus. Nevertheless, the mechanism of the signaling protein delivery to cilia is still poorly understood, especially for soluble proteins. To gain a better understanding of cytoplasmic proteins' transport to cilia we studied Gli transcription factors, large soluble proteins that accumulate at the tip of the cilium before their conversion into transcriptional activators[5,9,11].

Using proteomic screening, we found that Gli proteins interact with the exocyst, a complex implicated in ciliary delivery of membrane receptors[23,54]. We found that loss-of-function of the exocyst by RNAi, mitochondrial trap, or drug treatment decreases ciliary localization of the constitutively active mutant Gli2(P1-6A) independently of their effect on transmembrane Hh signaling proteins Ptch and Smo.

On a molecular level, we show that the N-terminal region of Gli proteins binds to the subcomplex I of the exocyst, which includes the Sec3/5/6/8 subunits[34,87]. This agrees with our data and published reports suggesting that the N-terminal domain is necessary for the Gli proteins ciliary accumulation[5,9,11]. The N-terminus is, however, not sufficient for Gli ciliary transport, with other domains, particularly the central domain of Gli2/3[5,9] likely participating in other stages of ciliary translocation, such as the passage through the diffusion barrier and the transport to the cilium tip.

Our results suggest that soluble cytoplasmic proteins, like Gli2/3, can use the exocyst as a vehicle for intracellular trafficking. The exocyst was shown to collaborate with the BLOC-1 complex and IFT20 in the transport of membrane proteins polycystin-2 and fibrocystin to cilia[23]. However, IFT20 does not interact with HA-Gli2(P1-6A) (Supplementary Fig. S4a). This suggests that the exocyst may mediate Gli protein ciliary trafficking independently of IFT20, which implies that the pathways directing membrane and soluble cilium components are somewhat divergent. Importantly, the exocyst can be transported to the cilium despite IFT20 loss-of-function[23].

Interestingly, we found that the exocyst subunits accumulate at the cilium base upon stimulation of the canonical Hh signaling. The most parsimonious explanation of this result is that the exocyst is co-transported with Gli proteins upon pathway activation. Alternatively, the exocyst might be delivered to the cilium base independently of Gli proteins, and then either trap these proteins at the base prior to their translocation to the cilium or promote their accumulation at the cilium indirectly. However, the fact that we see foci of interaction between Gli2 and the exocyst in areas distal to the cilium (Fig. 2b) suggests that the co-transport hypothesis is most likely.

Consistent with the requirement of the exocyst in the transport of Gli2 to cilia, it appears that Gli2 is associated, at least transiently, with intracellular vesicles. Interestingly, the subunits of the exocyst that most strongly interact with Gli2 are positioned away from the putative lipid-facing surface of the complex[34,87], indicating that the exocyst may form a tether between vesicle lipids and soluble proteins. Structural ciliary proteins had been previously found to be attached to the outer surfaces of intracellular vesicles carrying ciliary membrane proteins in *Chlamydomonas*[88]. We now provide functional data that corroborate and extend these findings. Protein delivery by vesicles to the cilium is persistent and essential for maintaining proper cilium function and structure[89,90]. Thus, the strategy of using vesicles as universal carriers of proteins, both soluble and membrane-embedded, to cilia, solves the logistical problem of homing many protein classes onto the tiny cilium base.

The trafficking of vesicles in cells is coordinated by the small GTPases from the Rab and Arf families. Intriguingly, we found

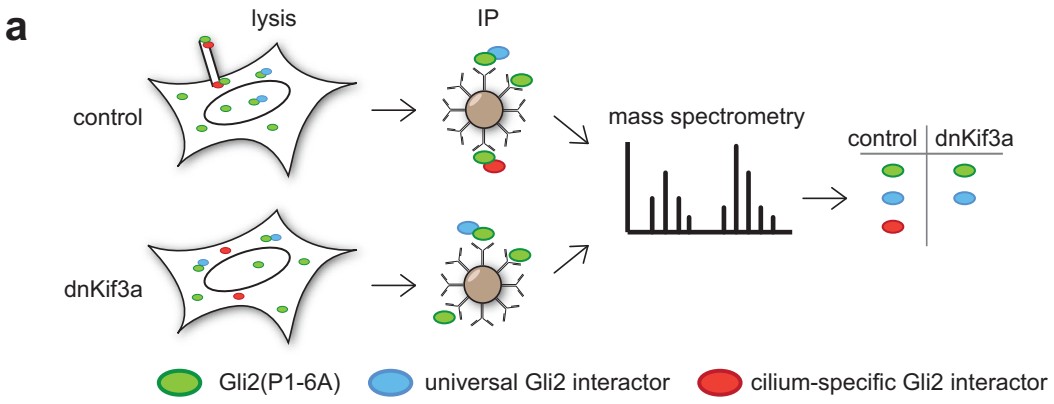

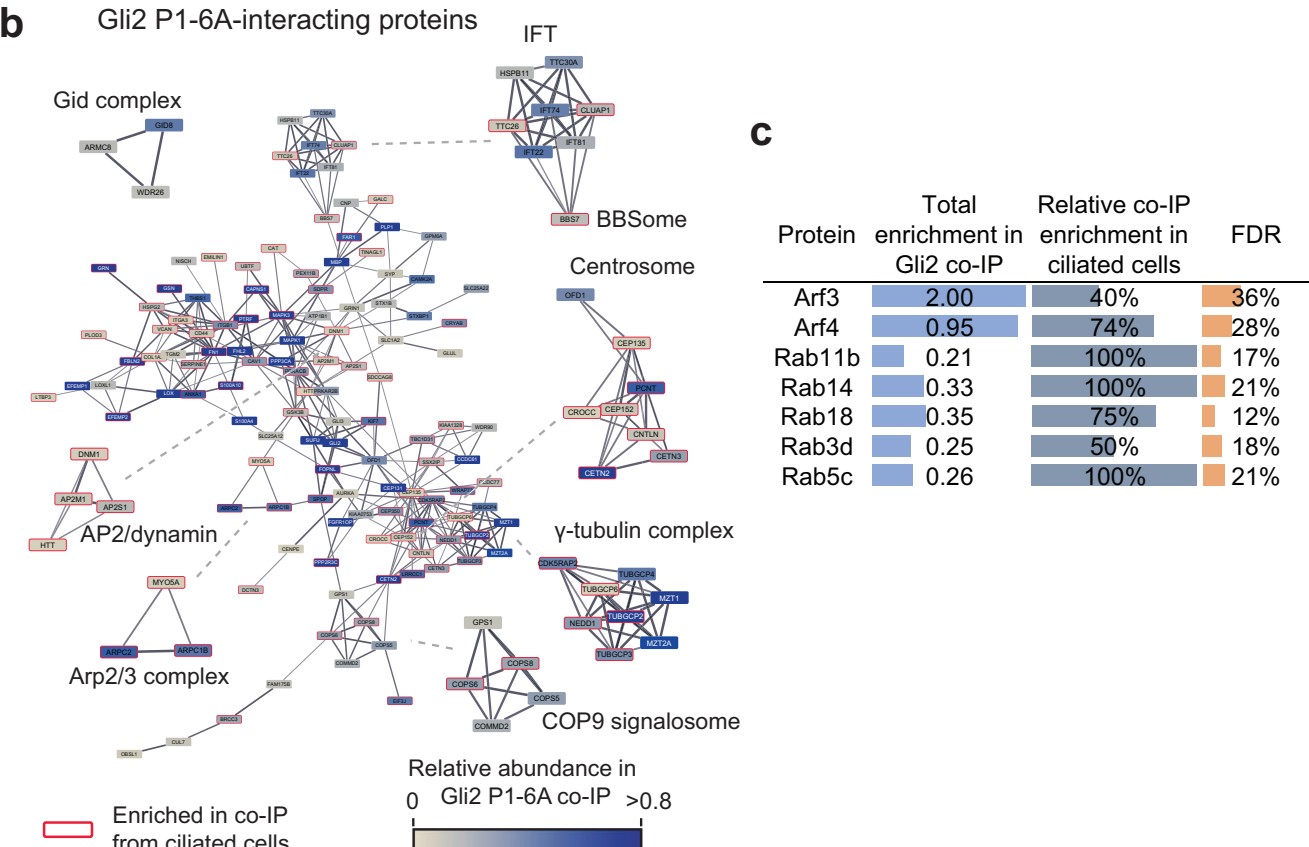

**Fig. 6 Interaction network of Gli2(P1-6A) in ciliated and non-ciliated cells. a** Schematic representation of the experiment. NIH/3T3 Flp-In cells stably expressing HA-Gli2(P1-6A) and either vector control or dominant-negative Kif3a (dnKif3a) were lysed in gentle lysis buffer and the lysates were immunoprecipitated using magnetic beads coated with anti-HA antibodies. Eluted proteins were submitted for mass spectrometric analysis. Common MS-AP contaminants (>10% FDR from the CRAPome database[49]) were removed from each dataset (control—ciliated, dnKif3a—non-ciliated). **b** High confidence HA-Gli2(P1-6A) interactors identified by MS were connected into a network using the STRING[118] plugin in Cytoscape. Proteins identified in Gli2(P1-6A) from ciliated and non-ciliated cells were pooled. Shown is the main protein network with the node color representing the approximate relative abundance of the protein in the Gli2 interactome and the edge thickness corresponding to the confidence of connection between proteins in the STRING database. Also shown are highly interconnected sub-networks identified using MCODE clustering, which typically corresponds to protein complexes or multiprotein functional units. Proteins that were identified predominantly in the ciliated cells are marked with red borders. **c** Small GTPases identified in Gli2(P1-6A) co-IP/MS experiments are shown, with their relative enrichment scores, relative enrichment in ciliated vs non-ciliated cell co-IP samples, and FDR scores based on the CRAPome database.

that Rab14, Rab18, Rab23, and Arf4, interact with Gli2 and are essential for its accumulation in the ciliary compartment. The Rab14 GTPase localizes at early endosomes and plays a role in protein exchange between the endosomes and the Golgi compartment[91–94], and exocytic vesicle targeting[95]. On the other hand, Rab18 is usually associated with the endoplasmic reticulum and lipid droplets[96,97]. Intriguingly, we identify COPI and TRAPP complex components in Gli2(P1-6A) and Gli3 inter-actomes, and these complexes have been implicated in lipid droplet recruitment of Rab18[98]. This suggests that Gli may recruit Rab18 via TRAPPII and COPI to promote ciliary trafficking. Interestingly, all three of the above GTPases: Rab18, Rab14, and

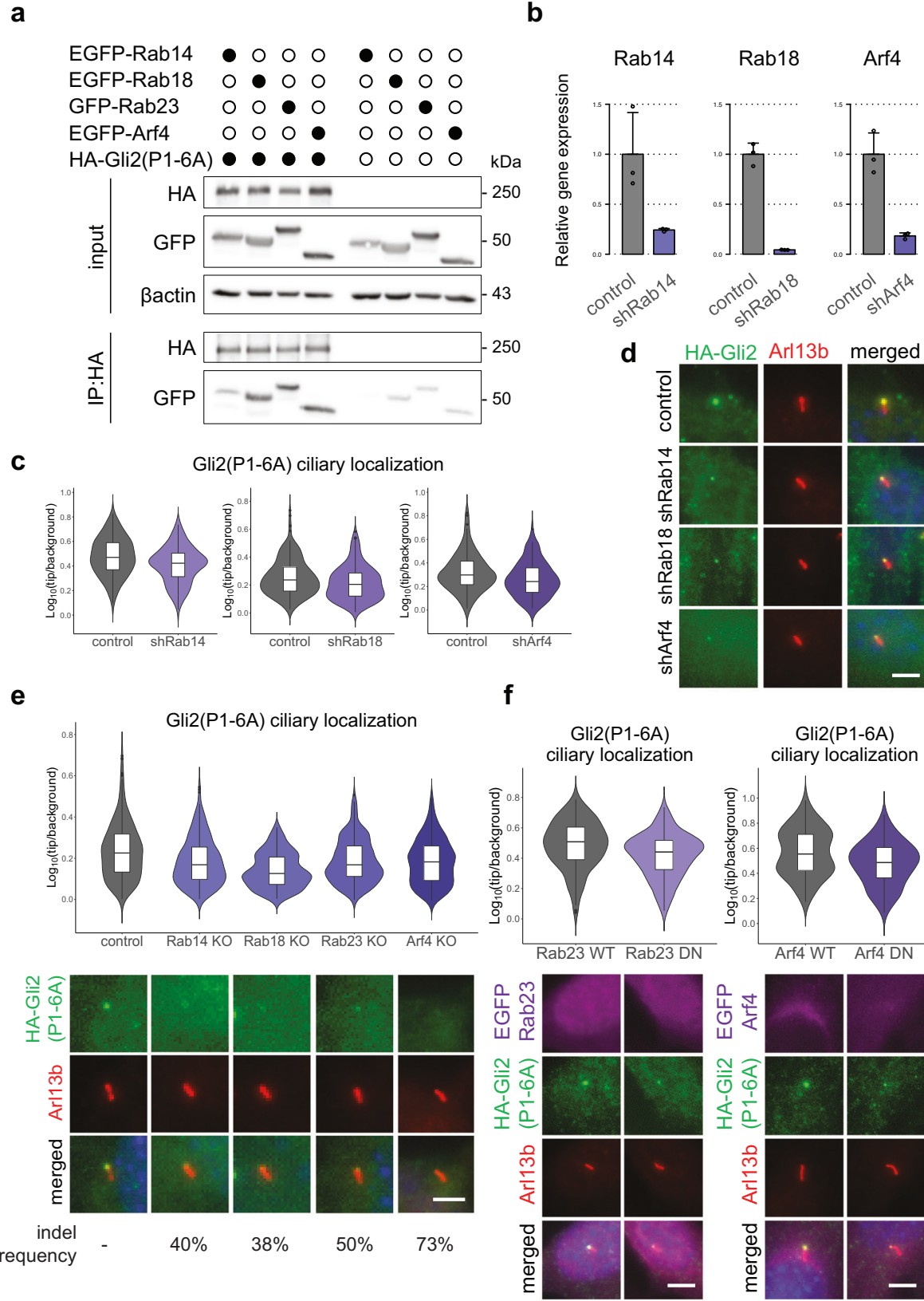

Arf4, were recently identified as proximity interactors of the cilium base-localized kinase Ttbk2[99], strengthening the case for their involvement in the targeting of Gli-laden vesicles to the cilium.

Finally, Rab23 had previously been implicated in Hh signaling and ciliary transport of receptors[100]. Rab23 is described as a

negative regulator of the Hh pathway but several different mechanisms have been proposed, from affecting Smo to directly regulating Gli proteins[75,76,101]. Here, we propose Rab23 as one of the key players in the trafficking of Gli transcription factors into the primary cilium. This is consistent with the recently discovered role of Rab23 in the transport of another soluble protein, Kif17, to

**Fig. 7 Rab14, Rab18, Rab23, and Arf4 mediate Gli2 ciliary trafficking into primary cilium. a** Co-immunoprecipitation of EGFP tagged Rab and Arf proteins with HA-Gli2(P1-6A). HEK293T cells were co-transfected with the indicated constructs and co-IP was performed as in Fig. 2c. **b** Knockdown efficiency of Rab14, Rab18, and Arf4 using shRNA. Cells were transduced with viral constructs encoding the indicated shRNAs and mRNA expression of their target genes was measured by qRT-PCR. Control cells were transduced with the shRNA against luciferase. **c** Effect of Rab14, Rab18, and Arf4 shRNA knockdown on relative Gli2(P1-6A) ciliary localization. Relative localization of Gli2(P1-6A) at the cilium tip was measured as in Fig. 3c for $n > 100$ cilia per group. Student's $t$ test control vs shRNA Rab14 $p$-value = 0.00018; control vs shRNA Rab18 $p$-value = 0.00027; control vs shRNA Arf4 $p$-value = 0.0081. **d** Representative images of HA-Gli2(P1-6A) localization in cilia of cells with the knockdown of Rab14, Rab18 and Arf4. Cells were transduced as in (**b**). Arl13B was used as a ciliary marker. **e** Effect of CRISPR-Cas9-mediated knockout of Rab14, Rab18, Rab23, and Arf4 on Gli2(P1-6A) ciliary localization. Cells stably expressing both HA-Gli2(P1-6A) and Cas9 were transduced with viral constructs encoding the indicated sgRNAs. Control cells were transduced with the empty pLentiGuide-puro vector. Relative localization of Gli2(P1-6A) at the cilium tip was measured as in Fig. 3c for $n > 280$ cilia per group. Student's $t$ test control vs Rab14 KO $p$-value = 1.1e−06; control vs Rab18 KO $p$-value < 2.2e−16; control vs Rab23 KO $p$-value = 3.4e−07; control vs Arf4 KO $p$-value = 8.9e−07. Representative images of Gli2(P1-6A) ciliary localization for each condition are presented below. Arl13b was used as a ciliary marker. Indel frequency measured using the TIDE method is shown below. **f** Effect of inducible expression of dominant-negative (DN) forms of Rab23 and Arf4 on Gli2(P1-6A) ciliary localization. Relative localization of Gli2(P1-6A) at the cilium tip was measured as in Fig. 3c for $n > 100$ cilia per group. Student's $t$ test Rab23 WT vs DN p-value = 2.2e−05; Arf4 WT vs DN $p$-value = 0.00031. Representative images of Gli2(P1-6A) ciliary localization for each condition are presented below. Arl13b was used as a ciliary marker. Scale bars 5 μm.

primary cilia and with the ciliary and early endosome enrichment of Rab23[102,103].

In addition to Rab family GTPases, we found Gli2 to associate with Arf4, which functions in sorting ciliary cargo at the Golgi and is a crucial regulator of ciliary receptor trafficking[104,105]. Arf4 binds the ciliary targeting signal of rhodopsin and controls the assembly of the Rab11a-Rabin8-Rab8 module for the proper delivery of cargo to the ciliary base[106]. Although Rab8 and Rab11a were found to cooperate with both the exocyst and Arf4[106] in the targeting of ciliary cargos, we found that the expression of dominant-negative Rab8 and Rab11a did not negatively affect Gli2 ciliary accumulation, with Rab8 DN actually promoting higher Gli2 accumulation in cilia (Supplementary Fig. S4c). Similarly, Rab8 and Rab11a KO did not reduce Gli2 ciliary trafficking (Supplementary Fig. S4b). Moreover, we did not find Rab8 or Rab11a among interactors of Gli2 and Gli3 in our co-IP/MS datasets. Instead, among Gli2 interactors was a Rab11a ortholog Rab11b, which had also been implicated in ciliogenesis and found to associate with Rab8[107,108]. Disentangling the roles of the two Rab11 orthologs as well as Rab8/Rabin8 in the trafficking of soluble ciliary components will be an interesting subject for future studies.

The data from loss of function studies of small GTPases did not allow us to identify a known source of vesicles that participated in Gli2 trafficking to cilia. Many of the implicated Rab/Arf proteins had been known to associate both with Golgi-derived exocytic vesicles and with plasma membrane-derived endosomes. To decipher the relative importance of these two potential vesicle sources, we used pharmacological inhibitors to show that Gli2 is likely delivered to cilia via endocytic vesicle trafficking rather than the canonical secretory pathway.

In addition to Gli2, other soluble ciliary proteins can adopt a similar transport mechanism. Specifically, we show that Lkb1 levels at primary cilia drop upon exocyst loss-of-function and inhibition of endocytosis. Like Gli2, Lkb1 associates with intracellular vesicles and interacts with the exocyst. Intriguingly, Lkb1 has been shown to be able to associate directly with plasma membrane phospholipids in addition to its nuclear and ciliary localization[109]. It will be interesting to determine if direct phospholipid binding is important for the targeting of Lkb1 to vesicles and its ciliary transport and whether phospholipid and exocyst association are synergistic. Incidentally, the exocyst and Lkb1 show similar specific affinity for phosphatidic acid[109,110]. It is tempting to speculate that other proteins that rely on vesicles for ciliary delivery, such as Gli2, may also show some affinity for membrane lipids even in the absence of exocyst tethering.

In contrast to Gli2 and Lkb1, another soluble ciliary component Ubxn10 localizes at the cilium normally in cells depleted of Sec5 or treated with dynasore. This suggests that while the vesicle-mediated transport is important for the ciliary localization of some cytoplasmic proteins, others use different routes of ciliary trafficking. Ubxn10 binds directly to the IFT-B complex via IFT38/CLUAP1[86], so it may be delivered to the primary cilium with the IFT particles. Alternatively, the dynamics of Ubxn10 exchange at the cilium may be slower than that of Gli2 or Lkb1, which prevents us from observing changes in its localization upon dynasore treatment. Nevertheless, the lack of effect of Sec5 knockdown on Ubxn10 ciliary accumulation argues against such an interpretation.

In summary, we describe a previously undocumented mechanism for the transport of soluble cytoplasmic proteins to primary cilia, which relies on the association of these proteins with dynamically cycling endocytic vesicles (Fig. 9). While we identify several key players in the ciliary trafficking of these vesicles, further work will dissect the precise sequence of events that are involved in this process. In particular, it will be interesting to discover potential similarities and differences between the canonical ciliary targeting pathways for membrane proteins, such as polycystin 2, fibrocystin, Smo, and rhodopsin with those described here for soluble ciliary proteins. Our work brings us closer to gaining a broad understanding of ciliary trafficking and the coordinated transport of proteins among membrane compartments.

## Methods

**Constructs and molecular cloning.** Gli2/3 constructs were cloned based on the Gli2(P1-6A) mutant previously described[55] tagged with the N-terminal 3xHA. Initially, Gli2 fragments were amplified by PCR and then cloned into the pENTR2B (Life Technologies) vector by Gibson assembly[111] using the NEBuilder® HiFi DNA Assembly Master Mix (NEB). Subsequently, the constructs were shuttled into pEF/FRT/V5-DEST (Life Technologies) using the Gateway method (Gateway LR Clonase II mix; Life Technologies). Plasmids with Sec3/5/8, Rab8/11/14/18, and Arf4 on the pEGFP vector were ordered from the Addgene site (Supplementary Table S1). Rab23 wild type and mutant cDNA sequences were obtained by DNA synthesis (DNA Strings; Thermo) and cloned by Gibson assembly into the LT3GEPIR plasmid ordered from addgene (Supplementary Table S1). Tom20 sequence was amplified from mouse cDNA and then fused with mScarlet cloned from pmScaret (addgene, Supplementary Table S1) and Sec5 by Gibson assembly in the pEGFP-C3 vector with the EGFP sequence removed by restriction digestion. Other

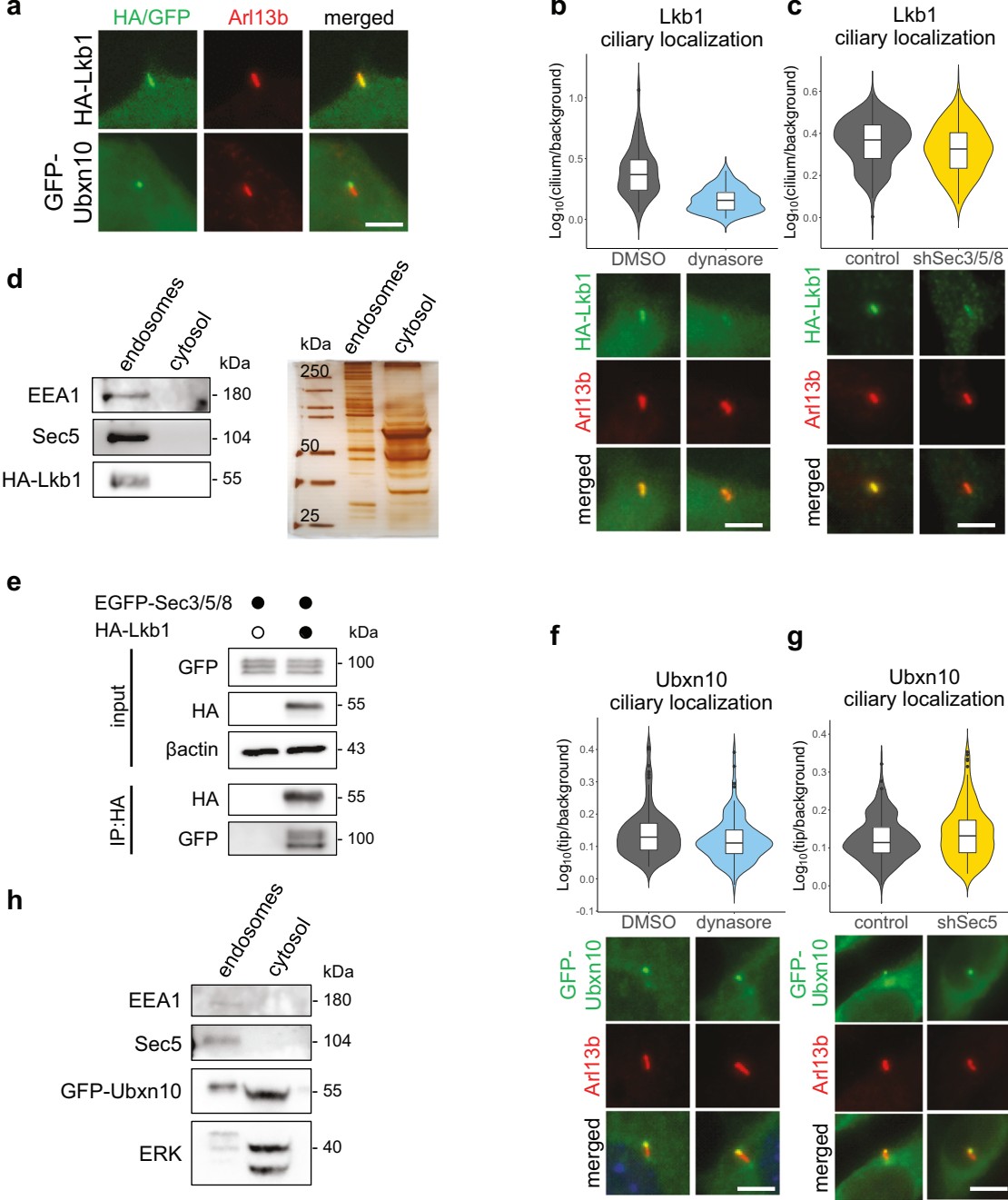

**Fig. 8 The trafficking of Lkb1, but not Ubxn10, depends on endocytosis and the exocyst. a** Ciliary localization of Lkb1 and Ubxn10 in NIH/3T3 cells. Cells were transfected and stained with the indicated antibodies. Arl13b was used as a ciliary marker. **b**, Effect of dynasore treatment on Lkb1 ciliary accumulation. NIH/3T3 cells with stable expression of HA-Lkb1 were treated with DMSO and dynasore (4h; 40μM). Relative localization of Lkb1 at the cilium was measured for n > 50 cells per group. Student's t test p-value = 2.3e−11. Representative images are presented below. **c** Effect of Sec3/5/8 shRNA knockdown on Lkb1 ciliary localization. Cells were transduced as in Fig. 3a. Relative localization of Lkb1 at the cilium tip was measured as in Fig. 3c for n > 70 cilia per group. Student's t test control vs shSec3/5/8 p-value = p-value = 0.003. **d** Cells stably expressing HA-Lkb1 were fractionated using the endosome isolation kit and the fractions were resolved using SDS-PAGE. Immunoblot shows Lkb1 in the endosomal fraction. EEA1 was used as a marker of the endosomes. Silver-stained gel of the same samples shows similar total protein abundance in both fractions. **e** Co-immunoprecipitation of EGFP tagged Sec3/5/8 proteins with HA-Lkb1 in HEK293T cells co-transfected with the indicated constructs. **f** Effect of dynasore treatment on Ubxn10 ciliary accumulation. Cells were treated as in (**b**) and relative localization was measured for n > 100 cells per group. Student's t test DMSO vs dynasore 4h p-value = 0.05. Representative images are presented below. **g** Effect of Sec5 shRNA knockdown on relative Ubxn10 ciliary localization. Cells were transduced as in Fig. 3a. Relative localization of Ubxn10 at the cilium tip was measured as in Fig. 3c for n > 160 cilia per group. Student's t test control vs shSec5 p-value = 0.037. **h** Fractionation of cells with stable expression of GFP-Ubxn10 as in (**d**). Immunoblot shows Ubxn10 mainly in the cytosolic fraction. ERK was used as a cytosolic fraction marker. Scale bars 5μm.

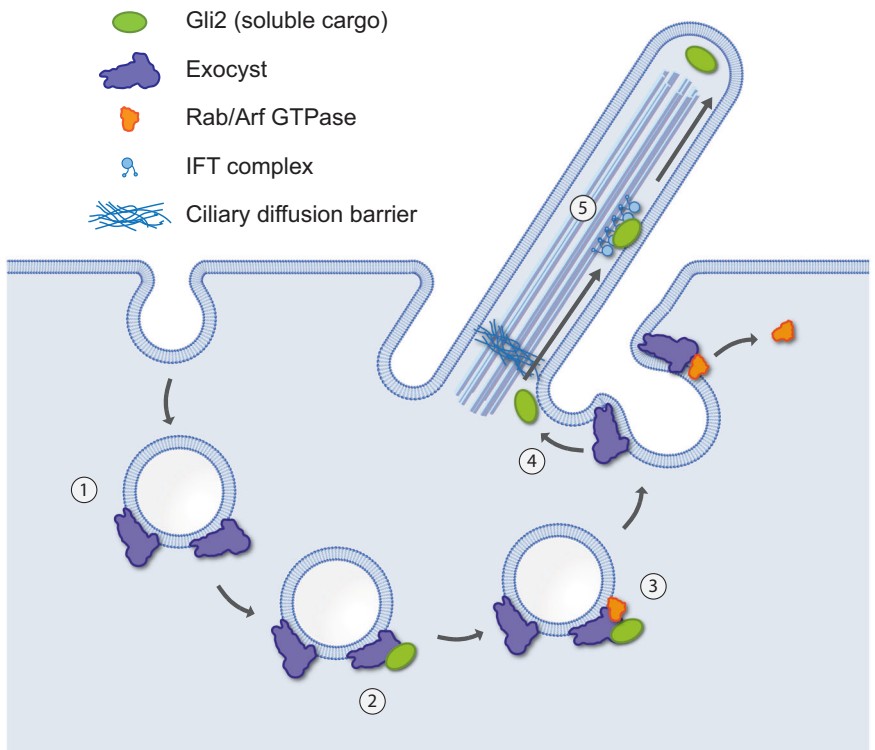

**Fig. 9 The model of Gli protein trafficking to cilia.** (1) Endocytic proteins that are targeted to cilia associate with the exocyst. (2) The exocyst subunits that face the cytoplasm capture soluble cargo. (3) Rab/Arf small GTPases associate with vesicles laden with soluble cargo and orchestrate transport to cilium base. (4) Vesicles dock at the ciliary pocket, and Gli proteins are released and transported across the ciliary barrier using importins. (5) Gli proteins are transported to cilium tip on IFT trains.

soluble proteins sequences of Dvl2, Nbr1, Mek1, Lkb1, Raptor were amplified from mouse cDNA and cloned into the pENTR2B with 3xHA tag vector by Gibson assembly. Ubnx10 was cloned from pHAGE-NGFP-UBXD3—gift from M. Raman[86]. Tbx3 was cloned from the construct with Tbx3-Myc—a gift from A. Moon[85]. pEGFP-Kap3a was a gift from P. Avasthi and pEGFP-Rab11a was a gift from M. Miaczynska. Primer sequences for cloning are shown in Supplementary Table S2.

**Cell culture**. HEK293T (ATCC) and NIH/3T3 Flp-In (Thermo) cells were maintained in media composed of DMEM (high glucose; Biowest), sodium pyruvate (Thermo), stable glutamine (Biowest), non-essential amino acids (Thermo), 10% fetal bovine serum (EurX), and penicillin/streptomycin solution (Thermo). mIMCD3 Flp-In cells (gift from D. Mick[112]) were cultured in DMEM/F-12 (Gibco) media supplemented with 2 mM stable glutamine (Biowest), 10% fetal bovine serum (EurX), and penicillin/streptomycin solution (Thermo). HA-Gli2(P1-6A) and HA-LKB1 NIH/3T3 and mIMCD3 stable cell lines were generated using the Flp-In system according to the manufacturer's protocols (Thermo Fisher). Stable cell lines were reselected with hygromycin on every other passage to preserve selection pressure.

To stimulate ciliogenesis the cells were cultured in the same medium but containing 0.5% FBS for 24h before fixing. For activation of the Hh pathway, we used SAG (Smoothened agonist) treatment 200nM for 24h. Transient transfections of cells we performed using the JetPrime reagent (Polyplus) according to the manufacturer's protocol.

All inhibitors were suspended in DMSO and used with indicated times. The following concentrations of inhibitors were used: dynasore (40μM, Sigma), endosidin2 (200μM, Sigma), pitstop2 (30μM, Sigma), brefeldin A (5μg/ml, Sigma).

**Large scale co-IP/MS on Gli3**. NIH/3T3 cells were cultured to confluence on 50 15cm dishes and starved overnight to promote ciliogenesis. They were treated with 100nM SAG for 4h. The cells were fractionated into "nuclear" and "cytoplasmic" fractions as previously described[113]. Briefly, cells were washed 2x with ice-cold PBS and 2x with ice-cold 10 mM HEPES pH 7.4. They were left on ice in 10 mM HEPES for 10 min. to swell. The HEPES buffer was removed and the cells were scraped in ice-cold SEAT (sucrose, EDTA, acetic acid, triethanolamine) buffer with protease and phosphatase inhibitors and homogenized using a Dounce homogenizer with nuclei release and integrity monitored microscopically. "Nuclear" and "cytoplasmic" fractions were separated using two rounds of centrifugation at 900xg for 5 min. at 4°C. The supernatant from the first centrifugation was collected as the "cytoplasmic" fraction, and the pellet from the first centrifugation was resuspended in SEAT buffer and centrifuged again. The pellet from the second centrifugation was collected as the "nuclear" fraction. The "cytoplasmic" fraction was supplemented with Tris (to a concentration of 50mM), NaCl (to a concentration of 150mM), Nonidet P-40 (to a concentration of 2%), sodium deoxycholate (to a concentration of 0.25%), DTT (to a concentration of 1mM), and protease inhibitors and incubated for 45 min. at 4°C. Afterwards, the "cytoplasmic" lysate was centrifuged at 20,000xg for 45 min, and the supernatant was used for immunoprecipitation. The "nuclear" pellet was lysed for 45 min in a buffer containing 50 mM Tris, 150 mM NaCl, 2% Nonidet P-40, 0.25 sodium deoxycholate, 1 mM DTT, and

protease and phosphatase inhibitors. The "nuclear" lysate was cleared by centrifugation at 20,000$g$ for 30 min. at 4 °C and used for immunoprecipitation. Each fraction was immunoprecipitated overnight with 150 μL Dynabeads-Protein G (Invitrogen) covalently cross-linked with goat-anti-Gli3 (AF3690; R&D Systems; 30 μg antibody per fraction). The beads were washed with the following buffers: harsh RIPA lysis buffer (50 mM Tris pH 7.4, 150 mM NaCl, 2% Nonidet P-40, 500 mM LiCl, 1 mM DTT, 0.25% sodium deoxycholate, 0.1% SDS, protease and phosphatase inhibitors), RIPA lysis buffer supplemented with 0.8M urea, and mild 0.1% NP-40 lysis buffer (50 mM Tris pH 7.4, 150 mM NaCl, 0.1% Nonidet P-40, 1 mM DTT, 1% glycerol, phosphatase inhibitors). The samples were eluted from beads using preheated 2x Laemmli sample buffer without DTT at 85 °C for 5 min. The samples were then reduced and alkylated using DTT and iodoacetamide and loaded onto a 6% SDS-PAGE gel. The gel was stained using the GelCode Blue reagent (Pierce) and prominent bands were excised using a sterile scalpel and submitted for further processing to MS Bioworks (Ann Arbor, MI). The bands were destained and subjected to in-gel digest using trypsin. Each gel digest was analyzed by nano LC/MS/MS with a Waters NanoAcquity HPLC system interfaced to a ThermoFisher LTQ Orbitrap Velos. Peptides were loaded on a trapping column and eluted over a 75μm analytical column at 350nL/min; both columns were packed with Jupiter Proteo resin (Phenomenex). The mass spectrometer was operated in data-dependent mode, with MS performed in the Orbitrap at 60,000 FWHM resolution and MS/MS performed in the LTQ. The fifteen most abundant ions were selected for MS/MS. Data were searched using a local copy of Mascot with the following parameters: Enzyme: Trypsin, Database: IPI Mouse v3.75 (forward and reverse appended with common contaminants), Fixed modification: Carbamidomethyl (C), Variable modifications: Oxidation (M), Acetyl (N-term, K), Pyro-Glu (N-term Q), Deamidation (N,Q), Phospho (S,T,Y), GlyGly (K), Mass values: Monoisotopic, Peptide Mass Tolerance: 10 ppm, Fragment Mass Tolerance: 0.5 Da, Max Missed Cleavages: 2. Mascot DAT files were parsed into the Scaffold algorithm for validation, filtering, and to create a nonredundant list per sample. Data were filtered using a minimum protein value of 90%, a minimum peptide value of 50% (Prophet scores), and requiring at least two unique peptides per protein.

To determine high-confidence Gli3 interactors, we rejected all proteins found in more than 10% of negative control affinity purification/MS experiments in the CRAPome database[49] (FDR < 10%). Enrichment of proteins representing specific Gene Ontology terms was performed using PANTHER with GO-Slim Cellular Component and GO-Slim Biological Process terms[114].

Peak files and protein identification results have been submitted to MassIVE (dataset MSV000093738 https://doi.org/10.25345/C5445HP7K).

**Large scale co-IP/MS on HA-Gli2 (P1-6A) in ciliated and non-ciliated cells.** NIH/3T3 cells stably expressing HA-Gli2 (P1-6A) were transduced either with the control vector or with a retroviral vector encoding the dominant-negative variant of Kif3a (headless – amino acids 441-701 of the mouse Kif3a; dnKif3a) and selected with puromycin to eliminate untransduced cells. Each cell line was expanded from a single clone and ciliogenesis or lack thereof was verified by immunofluorescence.

Both cell lines were starved for 36h and lysed in a gentle lysis buffer (50 mM Tris pH 7.4, 150 mM NaCl, 0.1% Nonidet P-40, 5% glycerol, protease and phosphatase inhibitors) and scraped at 4 °C. The lysate was clarified for 30 min at 15,000$g$ and the supernatant was immunoprecipitated for 2h at 4 °C with Dynabeads-protein G covalently coupled to the rat anti-HA

antibody (Roche). The beads were washed 3x5 min. with the lysis buffer and 1x5 min with the lysis buffer with the addition of 350 mM NaCl (total NaCl concentration 500mM). Protein was eluted from beads using 2x Laemmli sample buffer at 37 °C for 30 min with vigorous mixing (500rpm).

Eluted proteins were submitted for mass spectrometric protein identification to MS Bioworks (Ann Arbor, MI). The entire amount of sample was separated ~1.5 cm on a 10% Bis–Tris Novex mini-gel (Invitrogen) using the MES buffer system. The gels were stained with coomassie and excised into ten equally sized segments. Gel segments were processed using a robot (ProGest, DigiLab) with the following protocol: Washed with 25 mM ammonium bicarbonate followed by acetonitrile. Reduced with 10 mM dithiothreitol at 60 °C followed by alkylation with 50 mM iodoacetamide at RT. Digested with trypsin (Promega) at 37 °C for 4 h. Quenched with formic acid and the supernatant was analyzed directly without further processing.

The gel digests were analyzed by nano LC/MS/MS with a Waters M-class HPLC system interfaced with a ThermoFisher Fusion Lumos. Peptides were loaded on a trapping column and eluted over a 75μm analytical column at 350nL/min; both columns were packed with Luna C18 resin (Phenomenex). A 30 min gradient was employed (5h LC/MS/MS per sample). The mass spectrometer was operated in data-dependent mode, with MS and MS/MS performed in the Orbitrap at 60,000 FWHM resolution and 15,000 FWHM resolution, respectively. APD was turned on. The instrument was run with a 3s cycle for MS and MS/MS. Data were searched using a local copy of Mascot with the following parameters: Enzyme: Trypsin, Database: Swissprot Mouse (concatenated forward and reverse plus common contaminants), Fixed modification: Carbamidomethyl (C), Variable modifications: Oxidation (M), Acetyl (Protein N-term), Deamidation (NQ), Phosphorylation (S,T,Y), Mass values: Monoisotopic, Peptide Mass Tolerance: 10 ppm, Fragment Mass Tolerance: 0.02 Da, Max Missed Cleavages: 2. Mascot DAT files were parsed into the Scaffold software for validation, filtering, and to create a nonredundant list per sample. Data were filtered at 1% protein and peptide level FDR and requiring at least two unique peptides per protein.

Peak files and protein identification results have been submitted to MassIVE (dataset MSV000093739 https://doi.org/10.25345/C50G3H85P).

**Proteomic data analysis.** Proteomic data were analyzed using Scaffold 4 and Cytoscape 3.8.2 to generate and visualize protein-protein interaction networks.

**Viral transduction.** For lentivirus production, we transfected HEK293T cells with pRSV-rev, pMDLg/pRRE, pMD2.G lentiviral packaging vectors (addgene, Supplementary Table S1) and the construct encoding our protein or shRNA or sgRNA of interest, and then after 2 days, we collected the virus-containing medium and added it to target cells. We used puromycin to select transduced cells.

**siRNA mediated knockdown.** For siRNA-mediated knockdown of Sec5, we used the Sec5 ON-TARGET plus siRNA set of four siRNAs with non-targeting controls (Horizon Dharmacon). For siRNA transfection, we used Lipofectamine RNAiMAX (Thermofisher). Each siRNA was introduced at 40 pmol/well on a 24-well plate for 48h.

**shRNA mediated knockdown.** shRNAs were cloned into pLKO.1-TRC cloning vector (Supplementary Table S1). Targeting sequences were designed using the BlockIT software from the

Thermo-Fisher website. ShRNA primer sequences are in Supplementary Table S2.

**CRISPR-Cas9-mediated mutagenesis**. CRISPR-Cas9-mediated mutagenesis was performed on NIH/3T3 Flp-In cells stably expressing HA-Gli2(P1-6A) and Cas9 (Supplementary Table S1). sgRNA sequences were designed using the Broad Institute sgRNA designer tool[115] and cloned into the pLentiGuide-puro vector (addgene, Supplementary Table S1). We transduced the target cells with lentiviruses carrying the sgRNA of interest and either fixed 72 h later or subjected to antibiotic selection. Knockout efficiency was evaluated on pools of cells using Sanger sequencing and the TIDE method[116]. SgRNA primer sequences are in Supplementary Table S2.

**Immunostaining and microscopy**. Cells were cultured on glass coverslips. After low-serum starvation to promote ciliogenesis, we fixed cells in 4% [w/v] paraformaldehyde in PBS for 15 min at room temperature (RT) and then washed 3 x 10 min in phosphate buffer saline (PBS). Subsequently, cells were blocked and permeabilized in 5% [w/v] donkey serum in 0.2% [w/v] Triton X-100 in PBS. We incubated cells with the primary antibodies diluted in blocking buffer overnight at 4 °C. Next, we washed the coverslips 3 × 10 min with 0.05% [w/v] Triton X-100 in PBS, followed by incubation with secondary antibodies in the blocking buffer for 1 hour at RT. Cells were washed as above and mounted onto slides using a fluorescent mounting medium with DAPI (ProLong Diamond, Thermo). We acquired images on an inverted Olympus IX-73 fluorescent microscope equipped with a 63x uPLA-NAPO oil objective and the Photometrics Evolve 512 Delta camera using the cellSens sotware. For superresolution microscopy, we used the Zeiss LSM800 confocal microscope with the Airyscan detector and Plan Apochromat 63x/1.4 Oil DIC objective using the ZEN Black software.

For the quantitative analysis of fluorescence intensities, images were acquired with the same settings of exposure time, gain, offset, and illumination. Fluorescent intensities were measured in a semi-supervised manner by a custom ImageJ script. To calculate the Gli ciliary accumulation, we calculated the $\log_{10}$ values of the ratios of intensities of the fluorescent signal at the tip of the primary cilium and the surrounding background in each cell.

**Co-immunoprecipitation**. We performed co-immunoprecipitation using Pierce Anti-HA Magnetic Beads (Life Technologies) or using Dynabeads-protein G (Thermo) magnetic beads with primary antibodies (anti-GFP Genetex No#GTX113717; anti-Sec5 Proteintech No#12751-1-AP) crosslinked using dimethyl pimelimidate (Life Technologies).

For the production of whole-cell lysates, cells were lysed at 4 °C in lysis buffer (50 mM Tris at pH 7.4, 1% NP-40 [v/v], 150 mM NaCl, 0.25% sodium deoxycholate [v/v], protease inhibitor cocktail [1× EDTA-free protease inhibitors, Sigma], 10 mM NaF). 1/10 part of the clarified lysate was saved as an input fraction, and the rest was subjected to immunoprecipitation.

After adding beads, binding of the protein of interest was performed overnight with gentle rotation at 4 °C. The next day, beads were washed 4 × 10 min at 4 °C in the same lysis buffer to remove unbound proteins, and complexes were eluted off the beads using 2x SDS sample buffer at 37C for 30min. We analyzed the composition of eluent using the SDS-PAGE and Western Blot method.

**qRT-PCR**. Total RNA was isolated using the Universal RNA Purification Kit (EURx). Reverse transcription was performed using the High-Capacity cDNA Reverse Transcription Kit (Thermo). Quantitative PCR was performed on the Roche LightCycler 480 II system using primers shown in Supplementary Table S2 and the Real-Time 2xHS-PCR Master Mix Sybr B (A&A Biotechnology). Relative gene expression was quantified using the ΔΔCt method.

**SDS-PAGE and western blot**. Proteins were denatured for 30 min at 65 °C and resolved by SDS-PAGE. Afterward, we performed electrotransfer onto a nitrocellulose membrane. Immunocomplexes were detected using an enhanced chemiluminescence detection system (Clarity or Clarity Max, Bio-rad) on Amersham Imager 680 and 800 as 16-bit grayscale TIFF files. The molecular weight of proteins was estimated with pre-stained protein markers (Bio-rad).

**Proximity ligation assay**. We performed the proximity ligation assay[117] using the Duolink PLA Kit (Merck) according to the manufacturer's protocol. Anti-Sec5 and anti-Gli2 primary antibodies (Supplementary Table S3) were used to detect sites of interaction between the proteins in NIH/3T3 Flp-In cells.

**Endosome isolation**. The Trident Endosome Isolation Kit (Genetex) was used to fractionate cell lysates according to the manufacturer's protocol.

**Electron microscopy**. HEK293 cells expressing EGFP-Gli2(P-16A) were fixed on the dish with 4% PFA in 0.2M phosphate buffer and 0.25% sucrose. The samples were sent to Biocenter Oulu Electron Microscopy Core Facility and there processed for EM and immunogold labeled with anti-GFP. Imaging was performed on Sigma HD VP FE-SEM equipped with ET-SE and In-lens SE detectors, VPSE G3 detector for low vacuum mode, and 5Q-BSD detector.

**Transferrin uptake assay**. Cells were treated with DMSO or endocytosis inhibitors, followed by 20 min. incubation with 20µg/mL Alexa Fluor 647-conjugated transferrin (Sigma). The cells were washed with PBS, fixed with 4% PFA, and mounted in fluorescent mounting medium with DAPI (ProLong Diamond, Thermo).

**Statistics and reproducibility**. The statistical data analysis was performed using Microsoft Excel and R/RStudio (R version 4.1.2). For the processing of the fluorescence images, we used the FiJi/ImageJ suite. Statistical significance was calculated using Student's t test for experiments involving two experimental groups, or ANOVA and Tukey posthoc test for multiple comparisons. Number of samples (biological replicates) and p-value is provided in each figure legend. To ensure reproducibility in experiments where statistical analysis was not practical, each experiment was reproduced at least twice with results supporting the same conclusion.

## Data availability

All data supporting the findings of this study are available within the paper and its Supplementary Data. Raw data for plots are available as Supplementary Data 3. For coimmunoprecipitation/mass spectrometry experiments, data were submitted to the MassIVE database (accession code MSV000093738; https://doi.org/10.25345/C5445HP7K, accession code MSV000093739; https://doi.org/10.25345/C50G3H85P). Uncropped, unprocessed blot images from the Amersham Imager are included in Supplementary Fig. S6.

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

## Acknowledgements

The authors would like to thank Marta Miączyńska, Małgorzata Maksymowicz, Jarosław Cendrowski, and the members of the CeNT Bio PI discussion group, and the Laboratory of Molecular and Cellular Signalling for insightful discussion and helpful suggestions. We thank M. Raman, A. Moon, D. Mick, and P. Avasthi for sharing their reagents with us and Addgene contributors for making their plasmids available (see Supplementary Table S1). This work was supported by the following grants from the National Science Centre (NCN): OPUS 2021/43/B/NZ3/01457 and PRELUDIUM 2018/29/N/NZ3/01523. R.R. was supported by a grant from the National Institutes of Health (GM118082).

## Author contributions

S.N. conceived, planned, and executed most experiments, analyzed data, wrote the first draft of the manuscript, and participated in revisions, P.N. suggested the original idea, supervised the study, provided funding, helped with methods and reagents, and revised the manuscript, S.D., T.U., B.B., W.S. performed experiments, analyzed data, established methods, and provided reagents, E.W.H. and R.R. performed and analyzed the Gli3 co-IP/MS experiment and shared reagents.

## Competing interests

The authors declare no competing interests.
