## [Peer Review File · Communications Biology]

Reviewers' comments:

Reviewer #1 (Remarks to the Author):

The paper by Niedziolka et al, "The exocyst complex and intracellular vesicles 1 mediate soluble 2 protein trafficking to the primary cilium", addresses an important question, namely how proteins are trafficked to the primary cilia. This has important implications for organ function and cell signaling, especially via the Hedgehog pathway. Unfortunately, there are some serious concerns I have with the study design. The paper begins with a co-immunoprecipitation screen. Though it is not completely clear from the Methods, I believe the authors did the initial screen on NIH/3T3 cells stably expressing HA-Gli2 (P1-6A) as they describe in the Methods for the second "Large scale co-IP/MS on HA-Gli2 (P1-6A) in ciliated and non-ciliated cells". The legend for Figure 1 also notes that "NIH/3T3 Flp-In cells" were used. The problem with doing a co-immunoprecipitation screen using a highly overexpressed protein is that the results show what is possible when a protein is overexpressed but not necessarily what is actually going on in the cells. The choice of cell lines is also not clear to me as 3T3 cells do not have typical primary cilia that extend from the apical surface. Instead, in 3T3 cells the ciliary shaft is entirely enclosed in the cytoplasm of fully flattened cells (Albrecht-Buehler and Bushnell, *Experimental Cell Research*, 1980). The authors also confirm results in HEK293 cells (Figures 5 and 7). HEK293 cells contain minimal primary cilia and are not transcriptionally responsive to Hedgehog (Myers et al, *PNAS*, 2017). I would recommend using a typical ciliated cell line such as MDCK or IMCD3. The choice of the exocyst to study further, out of the many proteins identified in the screen, is a good choice but it has already been shown that the exocyst is necessary for trafficking proteins to the primary cilium. Indeed, when the Sec10 component of the exocyst was knocked down cilia didn't form though knockdown of Sec8 and Exo70 did not similarly inhibit ciliogenesis, which may be due to Sec10 being a central exocyst component, the absence of which results in the exocyst complex failing to assemble and being degraded (Zuo et al, *MBoC*, 2009). Likewise, the exocyst has already been shown to interact with Rab (Feng et al, *JBC*, 2012) and Arf (Seixis et al, *MBoC*, 2016) GTPases during ciliogenesis. I do think that the results using shRNA knockdown of exocyst components, a pharmacologic inhibitor of the exocyst (endosidin), and dynasore (though this could be non-specific) are more convincing; however, even if all the results are correct (which I am not convinced of), I think the results are more of an incremental advance.

Specific Concerns

1. In Figure 2, for the confirmation of the co-IP screen, I would have liked to see native proteins being used rather than overexpressed tagged proteins for the reasons detailed above. The 28 kDa GFP tag for exocyst proteins especially has been shown to interfere with localization and function. There are good antibodies that are commercially available for native exocyst proteins that work well for co-IP and IF (e.g. mouse monoclonal antibodies against Sec8 from Enzo). Also, in Figures 2D and 2F there is clearly a GFP band present, albeit at a lower level, in the absence of HA-Gli2 suggesting some non-specific interaction.
2. Again, in Figure 3 why not show localization of native Gli2 to the cilia rather than overexpressed HA-Gli2?
3. In Figure 7A there are again GFP bands seen when immunoprecipitation with HA is performed in the absence of any HA (HA-Gli2).

Reviewer #2 (Remarks to the Author):

The manuscript by Niedziolka et al. addresses the important issue of how soluble proteins are trafficked to cilia. It is carefully done and written, providing very interesting data and suggesting new hypotheses. However, there are several major and minor issues that must be addressed before the manuscript can be accepted for publication.

Major issues:

- 1 - The authors infer about localization at the ciliary tip/base without using specific markers. Therefore, at least a marker of the ciliary base should be added, so that the authors can take more robust conclusions.
- 2 - Arl13b is used throughout the manuscript to identify cilia, but it has been shown that mutations in EXOC2 cause defective Arl13b localization to the primary cilium (doi: 10.1084/jem.20192040). Therefore, acetylated tubulin should be used to validate that Arl13b localization to cilia is not affected when exocyst subunits are depleted (in Fig. 4E, acetylated tubulin staining is mentioned in the legend but not shown).
- 3 - On a related issue, why is Arl13b accumulated at the ciliary tip in Fig. 4B?
- 4 - In Fig. 5A, Sec5 appears inside the putative vesicles, as if it is a soluble cargo. Why does this occur? Thus, another exocyst subunit must be used to validate the result.
- 5 - In Fig. 5B, if the structures shown are vesicles, why are not any membranes visible? The authors have to show that there are membranes surrounding what they classify as vesicles.
- 6 - In Fig. 5C, other markers of endosomes must be used (e.g. Rab5, Rab7 and LAMP1), as well as markers of at least Golgi and ER, to guarantee that the authors are analyzing endosomes only and clarify what type of endosomes are.
- 7 - Controls for the inhibition of endocytosis by dynasore and pitstop are missing (showing that endocytosis was actually impaired and by what degree).
- 8 - The authors chose not to study Rab8 or Rab11a because they "do not strongly bind to Gli2". However, in Fig. S4, the binding to Rab11a is quite strong. What was the threshold/criterion used? Then, the authors refer that there are no differences in Gli2 ciliary localization upon expression of Rab11a or Rab8 DN mutants. Did the authors try to deplete these proteins to see if the same occurs? This should be clarified.
- 9 - On a related point, Rabs and Arfs are mostly/only membrane-bound when they are in their active form. Therefore, to validate that the interactions with Rab14, Rab18, Rab23 and Arf4 occur in vesicles, the constitutively active forms of the Rabs/Arf should be used. The DN forms could serve as a control. And why do the authors did not perform the KO of Arf4 and only use DN for Rab23 and Arf4, and not also for Rab14 and Rab23? This should be added.
- 10 - The authors should be more clear and discuss better what is the model they propose. Is it that exocyst tethers Gli2 to vesicles? And if endocytosis was shown to be required (Fig. 5), do the authors favor the route of delivery to the plasma membrane and then endocytosis and recycling to the cilium? This must be clarified.

Minor issues:

- 1 - Scale bars are missing in all figures.
- 2 - Stats should be shown in all the plots (with stars).
- 3 - In Fig. 8, the immunofluorescence images corresponding to the plots shown in B and C should be displayed, like in the rest of the manuscript.

Reviewer #3 (Remarks to the Author):

In this manuscript, Niedziółka SM et al report a novel role of the exocyst in the ciliary trafficking of soluble proteins. Roles of the exocyst complex in ciliary trafficking of transmembrane compartments have been well studied. How soluble proteins including Gli transcription factors are delivered to the primary cilium is less clear. Here the authors show that the exocyst interacts with Gli proteins by mass spec study. They further proved that the accumulation of Gli proteins in cilium relies on the exocyst-mediated vesicle trafficking to the ciliary base. This work proposes a novel mechanism of soluble protein transport from cytosol to the cilium.

previous studies showed that the exocyst proteins were also important for ciliogenesis. Thus, it is critical to monitor the cilia frequency and length in the exocyst protein depleted cells to ensure the reduction of Gli accumulation in cilia is caused by trafficking but not cilia assembly defects. Do the authors have data on cilia counting in Sec proteins depleted cells?

1. In the PLA assay, how often was the interaction detected on the base of the cilia?
2. Fig4A, does Gli2 localize to mitochondrial with sec5-Tom20 fusion protein?
3. In Fig5A, the bright dots seem too big to be vesicles. What are the sizes of these dots?
4. The authors carried out CRISPR knockout of Rab14, Rab18, and Rab23. Are these pools of cells or single colonies? The KO efficiency needs to be validated with antibodies or Sanger sequencing.
5. Scale bars are missing for most of the fluorescent images.

We would like to thank the reviewers for their insightful comments on our manuscript. We have now modified our manuscript and include extensive new data to address the reviewers' concerns (Fig. S1A, S2A, S2B, S2D, S2E, S3A, S3C, S3E, S3F, S3G, S4C, new data in Fig. 7E, and Fig. 4B). Please see below for our detailed responses

Reviewer #1:

[...] The paper begins with a co-immunoprecipitation screen. Though it is not completely clear from the Methods, I believe the authors did the initial screen on NIH/3T3 cells stably expressing HA-Gli2 (P1-6A) as they describe in the Methods for the second "Large scale co-IP/MS on HA-Gli2 (P1-6A) in ciliated and non-ciliated cells". The legend for Figure 1 also notes that "NIH/3T3 Flp-In cells" were used. The problem with doing a co-immunoprecipitation screen using a highly overexpressed protein is that the results show what is possible when a protein is overexpressed but not necessarily what is actually going on in the cells.

The initial co-IP screen was done using endogenous Gli3 as bait (see section 'Large scale co-IP/MS on Gli3' in Methods. We performed confirmatory co-IP/WB experiments using Sec5 or Sec3 as bait and either endogenous Gli2 and Gli3 or HA-tagged Gli2 or Gli3 stably expressed at near-endogenous levels using the Flp-In system as prey (see Fig. 2A and newly added Fig. S1A). Neither of these experiments was performed using overexpressed proteins, which makes us confident that the results are physiologically relevant.

The choice of cell lines is also not clear to me as 3T3 cells do not have typical primary cilia that extend from the apical surface. Instead, in 3T3 cells the ciliary shaft is entirely enclosed in the cytoplasm of fully flattened cells (Albrecht-Buehler and Bushnell, Experimental Cell Research, 1980).

NIH/3T3 cells are a model that's often used in the study of Hh signaling, respond well to Hh pathway stimulation, and express Gli proteins. The fact that the cilia are located inside deep ciliary pockets in some NIH/3T3 cells does not change the membrane topology of the cilium. The ciliary base remains the only point at which a soluble protein can enter the cilium to reach the tip (see Fig. 3 in Albrecht-Buehler and Bushnell, 1980). Therefore, they are good models to study the transport of soluble proteins to the primary cilium. Importantly, our key results were reproduced in a different cellular model – mIMCD3, considered to be one of the primary models of ciliogenesis (see Fig. S2B and S3F, G).

The authors also confirm results in HEK293 cells (Figures 5 and 7). HEK293 cells contain minimal primary cilia and are not transcriptionally responsive to Hedgehog (Myers et al, PNAS, 2017). I would recommend using a typical ciliated cell line such as MDCK or IMCD3.

In our hands, HEK293 extend primary cilia and accumulate Gli proteins at cilia tips (see newly added Fig. S2E). While they don't respond to the Hh ligand or Smo agonist, they are capable of responding transcriptionally to Gli protein overexpression, as demonstrated by luciferase reporter assays. One of the reasons why Myers et al. may not have observed primary cilia in

HEK293 cells is that these cells require prolonged serum starvation to block proliferation and switch to the G0 phase and extend cilia.

As mentioned above, in accordance with the reviewer's recommendation we also used mIMCD3 cells and reproduced the key results in this more 'traditional' ciliogenesis model.

The choice of the exocyst to study further, out of the many proteins identified in the screen, is a good choice but it has already been shown that the exocyst is necessary for trafficking proteins to the primary cilium. Indeed, when the Sec10 component of the exocyst was knocked down cilia didn't form though knockdown of Sec8 and Exo70 did not similarly inhibit ciliogenesis, which may be due to Sec10 being a central exocyst component, the absence of which results in the exocyst complex failing to assemble and being degraded (Zuo et al, MBoC, 2009).

Likewise, the exocyst has already been shown to interact with Rab (Feng et al, JBC, 2012) and Arf (Seixis et al, MBoC, 2016) GTPases during ciliogenesis. I do think that the results using shRNA knockdown of exocyst components, a pharmacologic inhibitor of the exocyst (endosidin), and dynasore (though this could be non-specific) are more convincing; however, even if all the results are correct (which I am not convinced of), I think the results are more of an incremental advance.

Indeed, we cite all of the above papers in our manuscript. We don't believe that these prior publications diminish the novelty of our paper. They are mostly focused on the broad effects of exocyst on ciliogenesis (Zuo et al.) or on the physical interactions between the exocyst and small GTPases (Seixis et al., and Feng et al.). Our novel finding is that soluble ciliary proteins, for which no known molecular mechanism of ciliary targeting had been discovered, use the exocyst and vesicle transport for their delivery to the cilium. Importantly, in our hands the exocyst loss of function does not produce a statistically significant reduction in the number of ciliated cells (newly added Fig. S2D).

Specific Concerns

1. In Figure 2, for the confirmation of the co-IP screen, I would have liked to see native proteins being used rather than overexpressed tagged proteins for the reasons detailed above. The 28 kDa GFP tag for exocyst proteins especially has been shown to interfere with localization and function. There are good antibodies that are commercially available for native exocyst proteins that work well for co-IP and IF (e.g. mouse monoclonal antibodies against Sec8 from Enzo). Also, in Figures 2D and 2F there is clearly a GFP band present, albeit at a lower level, in the absence of HA-Gli2 suggesting some non-specific interaction.

Fig. 2A and the newly added Fig. S1A show interaction between endogenous Sec5 and Sec3 and the endogenous Gli2, Gli3, and stably expressed near-endogenous levels of HA-Gli2 and HA-Gli3. Background levels of prey proteins are typically present in eluates of almost all co-IP experiments due to non-specific interactions of preys with the beads. However, there is significantly more prey eluted from bait-laden beads than from beads without the bait, which demonstrates the specificity of the interaction.

2. Again, in Figure 3 why not show localization of native Gli2 to the cilia rather than overexpressed HA-Gli2?

We would like to stress that the level of stably expressed Gli2 and Gli3 in Flp-in cells are near-endogenous (see Niewiadomski et al., 2014). The two primary reasons why we used stably expressed HA-Gli2 P1-6A in most of our experiments was (1) because Gli2 P1-6A (constitutively active mutant) localizes to cilia independently of Smoothed and thus its localization would not be affected if our loss of function manipulations affected Smoothed or Patched localization and (2) because HA-tagged Gli proteins can be detected with high affinity anti-HA antibodies which produce high signal-to-background ratios.

3. In Figure 7A there are again GFP bands seen when immunoprecipitation with HA is performed in the absence of any HA (HA-Gli2).

See response to remark 1.

Reviewer #2 (Remarks to the Author):

The manuscript by Niedziolka et al. addresses the important issue of how soluble proteins are trafficked to cilia. It is carefully done and written, providing very interesting data and suggesting new hypotheses. However, there are several major and minor issues that must be addressed before the manuscript can be accepted for publication.

Major issues:

1 - The authors infer about localization at the ciliary tip/base without using specific markers. Therefore, at least a marker of the ciliary base should be added, so that the authors can take more robust conclusions.

We have now included a representative image with pericentrin used as the base marker (newly added Fig. S2A). Overall our inference was based on extensive experience we have imaging Gli proteins in cells where only the tip of the cilium shows strong accumulation of these proteins.

2 - Arl13b is used throughout the manuscript to identify cilia, but it has been shown that mutations in EXOC2 cause defective Arl13b localization to the primary cilium (doi: 10.1084/jem.20192040). Therefore, acetylated tubulin should be used to validate that Arl13b localization to cilia is not affected when exocyst subunits are depleted (in Fig. 4E, acetylated tubulin staining is mentioned in the legend but not shown).

Our method of measuring ciliary accumulation of Gli proteins does not depend on levels of Arl13b in cilia. Arl13b is simply used to define the geometry of the cilium and any signal above a certain threshold is considered to belong to the ciliary compartment. The cilia are then 'skeletonized' to remove artifacts related to different apparent thickness, and tip/base are identified as constant radius circles located at the ends of the skeletonized rod. We do not compare the intensity of Gli proteins to the intensity of Arl13b signal – if the Arl13b signal is below threshold the cilium is simply rejected as a whole from the calculation. We don't observe

a reduction in the number of Arl13b-positive cilia identified in exocyst-depleted cells (see above, Fig. S2D), suggesting that our method of cilia identification using Arl13b staining is not impaired under these conditions. Moreover, we performed additional experiments to compare Gli2(P1-6A) localization in dynasore-treated cells co-stained with anti-HA and anti-acetylated tubulin (an alternative cilia marker) and obtained results similar to those where we used Arl13b as a marker (newly added Fig. S3E)

3 - On a related issue, why is Arl13b accumulated at the ciliary tip in Fig. 4B?

We tend to observe massive accumulation of Gli proteins in tips HEK293 cells, which results in an Arl13b-positive 'bulge' to form at the tip. It doesn't seem like it's an actual accumulation of Arl13b, but rather a geometrically larger area that is Arl13b-positive.

4 - In Fig. 5A, Sec5 appears inside the putative vesicles, as if it is a soluble cargo. Why does this occur? Thus, another exocyst subunit must be used to validate the result.

We obtained very similar results using different Sec subunits. Our hypothesis is that the resolution of our microscope is not sufficient to show hollow structures of sub-micron size. We attempted to image our samples with superresolution microscopy, but were unable to obtain images of sufficient quality.

5 - In Fig. 5B, if the structures shown are vesicles, why are not any membranes visible? The authors have to show that there are membranes surrounding what they classify as vesicles.

The Immuno-EM technique used in this experiment results in poor contrast of intracellular membrane structures. However, the round pattern of Gli2 immuno-EM signal, when combined with other pieces of evidence, suggests that the protein accumulates around vesicles.

6 - In Fig. 5C, other markers of endosomes must be used (e.g. Rab5, Rab7 and LAMP1), as well as markers of at least Golgi and ER, to guarantee that the authors are analyzing endosomes only and clarify what type of endosomes are.

We have now performed additional western blots using markers of lysosomes (Rab7) and ER (Calnexin). We indeed see some calnexin staining in the endosomal fraction, suggesting that a partial contamination with the ER is present in the kit-isolated endosomes. However, Rab7 is excluded from the endosomal fraction, suggesting that lysosomes are not co-purified with endosomes using this method (Fig. S3A). For Rab5 and LAMP1 we did not detect any signal on western blot in either fraction, suggesting that the antibodies we used are not suitable for blotting. However, the co-IP experiments we performed between Gli2 and Rab14, Rab18, Rab23, and Arf4 (Fig. 7) provide additional proof of the physical interaction between Gli2 and endosomes.

7 - Controls for the inhibition of endocytosis by dynasore and pitstop are missing (showing that endocytosis was actually impaired and by what degree).

We have now performed experiments showing the efficacy of dynasore and pitstop2 at reducing transferrin uptake, which is a standard method of measuring endocytosis rate. Both compounds reduce the uptake of transferrin at the times and doses that result in a reduction of Gli2 ciliary accumulation (Fig. S3C)

8 - The authors chose not to study Rab8 or Rab11a because they "do not strongly bind to Gli2". However, in Fig. S4, the binding to Rab11a is quite strong. What was the threshold/criterion used? Then, the authors refer that there are no differences in Gli2 ciliary localization upon expression of Rab11a or Rab8 DN mutants. Did the authors try to deplete these proteins to see if the same occurs? This should be clarified.

We depleted Rab11a and Rab8 using 2 different sgRNAs each. Depending on the sgRNA used we either observed a small increase in Gli2(P1-6A) ciliary localization or no effect (newly added Fig. S4C), which is consistent with experiments performed using dominant negative Rab11a and Rab8 mutants (Fig. S4B) and further demonstrates that these two Rab GTPases are not required for Gli2 ciliary accumulation.

9 - On a related point, Rabs and Arfs are mostly/only membrane-bound when they are in their active form. Therefore, to validate that the interactions with Rab14, Rab18, Rab23 and Arf4 occur in vesicles, the constitutively active forms of the Rabs/Arf should be used. The DN forms could serve as a control. And why do the authors did not perform the KO of Arf4 and only use DN for Rab23 and Arf4, and not also for Rab14 and Rab23? This should be added.

We do not claim that the interaction occurs exclusively on vesicles. The functional effect of dominant negative GTPases suggest that active GTPase isoforms are necessary for the targeting of Gli-laden vesicles to the cilium, but not that the interaction is limited to vesicle-bound Gli proteins. We now include Arf4 KO results in Fig. 7E. The reason why they were not initially included in the figure was that we could not, for technical reasons, demonstrate the efficiency of Arf4 KO. Following our initial submission, we solved the technical issues and showed >70% KO efficiency for Arf4. For Rab14 and Rab18, we were unable to obtain detectable overexpression of the dominant negative mutants, so we made our conclusions based on sgRNA and shRNA-based loss of function.

10 - The authors should be more clear and discuss better what is the model they propose. Is it that exocyst tethers Gli2 to vesicles? And if endocytosis was shown to be required (Fig. 5), do the authors favor the route of delivery to the plasma membrane and then endocytosis and recycling to the cilium? This must be clarified.

The model has been added as Fig. 9. Our hypothesis is that the exocyst tethers Gli proteins to the vesicles, which attracts cilium-targeting Rab proteins.

Minor issues:

1 - Scale bars are missing in all figures.

We now include scale bars in all fluorescence panels.

2 - Stats should be shown in all the plots (with stars).

We include p-values in figure legends. Marking significance as 'stars' is a discouraged practice that Nature Publishing Group journals specifically forbid.

3 - In Fig. 8, the immunofluorescence images corresponding to the plots shown in B and C should be displayed, like in the rest of the manuscript.

The immunofluorescence images have been added to the figure.

Reviewer #3 (Remarks to the Author):

In this manuscript, Niedziółka SM et al report a novel role of the exocyst in the ciliary trafficking of soluble proteins. Roles of the exocyst complex in ciliary trafficking of transmembrane compartments have been well studied. How soluble proteins including Gli transcription factors are delivered to the primary cilium is less clear. Here the authors show that the exocyst interacts with Gli proteins by mass spec study. They further proved that the accumulation of Gli proteins in cilium relies on the exocyst-mediated vesicle trafficking to the ciliary base. This work proposes a novel mechanism of soluble protein transport from cytosol to the cilium.

previous studies showed that the exocyst proteins were also important for ciliogenesis. Thus, it is critical to monitor the cilia frequency and length in the exocyst protein depleted cells to ensure the reduction of Gli accumulation in cilia is caused by trafficking but not cilia assembly defects. Do the authors have data on cilia counting in Sec proteins depleted cells?

We counted cilia numbers in exocyst-depleted cells and did not detect any reduction in cilia numbers (see newly added Fig. S2D), suggesting that ciliogenesis is not affected under conditions we applied in our experiments. We also did not detect any reduction in cilia length in exocyst-depleted cells (not shown).

1. In the PLA assay, how often was the interaction detected on the base of the cilia?

The ciliary localization of the interaction was seen infrequently. We believe that the reason is that at the cilium base is actually where the interaction dissolves and Gli proteins enter the cilium (see newly added Fig. 9). Our model is that the exocyst tethers Gli proteins to vesicles in transit, hence the predominant localization of the interaction spots throughout the cell.

2. Fig4A, does Gli2 localize to mitochondrial with sec5-Tom20 fusion protein?

Yes. Please see the newly added panels on the right in Fig. 4B.

3. In Fig5A, the bright dots seem too big to be vesicles. What are the sizes of these dots?

We believe that under conditions of overexpression, some Gli2-laden vesicles probably fuse to form larger vesicles. However, quite a few of the dots are considerably smaller.

4. The authors carried out CRISPR knockout of Rab14, Rab18, and Rab23. Are these pools of cells or single colonies? The KO efficiency needs to be validated with antibodies or Sanger sequencing.

These were done on pools of cells. We have now included results of TIDE analysis of sequencing of pooled genomic DNA for each sgRNA in Fig. 7E.

5. Scale bars are missing for most of the fluorescent images.

We now include scale bars in all fluorescence panels.

Reviewers' comments:

Reviewer #2 (Remarks to the Author):

The authors addressed most of the reviewers' concerns and provide new (although limited) data. Moreover, they now provide a schematic with the working model, which is very useful. However, the authors do not provide:

1 - Evidence obtained with another marker (e.g., acetylated tubulin) that the number of cilia (rather than Arl13b-positive cilia) is not affected in exocyst-depleted cells;

2 - Data obtained with another exocyst subunit to solve the issue raised in point 4, since they could not provide super-resolution images for Sec5;

3 - Another marker of early endosomes (e.g., EEA1), since Rab7 is not a marker of lysosomes (rather, late endosomes; this should be corrected in the manuscript), suggesting that the endosomes identified are early endosomes.

These seem to be straightforward experiments that do not require a long time to perform and that would make this a stronger paper.

Reviewer #3 (Remarks to the Author):

The authors have addressed most of my concerns and should be published.

We thank the reviewer for their valuable input. We have addressed their remarks in the manuscript and figures, as detailed below.

1 - Evidence obtained with another marker (e.g., acetylated tubulin) that the number of cilia (rather than Arl13b-positive cilia) is not affected in exocyst-depleted cells;

We have now performed the quantification using acetylated tubulin as marker of primary cilia and the results are shown in Fig. S2D. The knockdown of either of the three exocyst subunits did not cause a significant change in the fraction of cells with acetylated tubulin-positive cilia, mirroring the result obtained with Arl13b as the cilia marker.

2 - Data obtained with another exocyst subunit to solve the issue raised in point 4, since they could not provide super-resolution images for Sec5;

We have now performed co-immunofluorescence imaging of Sec3 and Gli2. While the Sec3-positive structures are less obviously vesicular, we observe very strong colocalization of Gli2 and Sec3 throughout the cell (Fig. 5A).

3 - Another marker of early endosomes (e.g., EEA1), since Rab7 is not a marker of lysosomes (rather, late endosomes; this should be corrected in the manuscript), suggesting that the endosomes identified are early endosomes.

We have now corrected the manuscript to describe Rab7 as the marker of late endosomes. EEA1 is included as a marker of early endosomes in Fig. 5C and co-fractionates with Sec5 and Gli2/3.